# Generalized Matrix Means for Semi-Supervised Learning with Multilayer Graphs

**Pedro Mercado[1], Francesco Tudisco[2] and Matthias Hein[1]**
[1]University of Tübingen, Germany
[2]Gran Sasso Science Institute, Italy

## Abstract

We study the task of semi-supervised learning on multilayer graphs by taking into account both labeled and unlabeled observations together with the information encoded by each individual graph layer. We propose a regularizer based on the generalized matrix mean, which is a one-parameter family of matrix means that includes the arithmetic, geometric and harmonic means as particular cases. We analyze it in expectation under a Multilayer Stochastic Block Model and verify numerically that it outperforms state of the art methods. Moreover, we introduce a matrix-free numerical scheme based on contour integral quadratures and Krylov subspace solvers that scales to large sparse multilayer graphs.

## 1 Introduction

The task of graph-based Semi-Supervised Learning (SSL) is to build a classifier that takes into account both labeled and unlabeled observations, together with the information encoded by a given graph[4, 27]. A common and successful approach is to take a suitable loss function on the labeled nodes and a regularizer which provides information encoded by the graph [2, 15, 30, 32, 35]. Whereas this task is well studied, traditionally these methods assume that the graph is composed by interactions of one single kind, i.e. only one graph is available.

For the case where multiple graphs, or equivalently, multiple layers are available, the challenge is to boost the classification performance by merging the information encoded in each graph. The arguably most popular approach for this task consists of finding some form of convex combination of graph matrices, where more informative graphs receive a larger weight [1, 13, 14, 23, 28, 29, 31, 33].

Note that a convex combination of graph matrices can be seen as a weighted arithmetic mean of graph matrices. In the context of multilayer graph clustering, previous studies [19–21] have shown that weighted arithmetic means are suboptimal under certain benchmark generative graph models, whereas other matrix means, such as the geometric [20] and harmonic means [19], are able to discover clustering structures that the arithmetic means overlook.

In this paper we study the task of semi-supervised learning with multilayer graphs with a novel regularizer based on the power mean Laplacian. The power mean Laplacian is a one-parameter family of Laplacian matrix means that includes as special cases the arithmetic, geometric and harmonic mean of Laplacian matrices.We show that in expectation under a Multilayer Stochastic Block Model, our approach provably correctly classifies unlabeled nodes in settings where state of the art approaches fail. In particular, a limit case of our method is provably robust against noise, yielding good classification performance as long as one layer is informative and remaining layers are potentially just noise. We verify the analysis in expectation with extensive experiments with random graphs, showing that our approach compares favorably with state of the art methods, yielding a good classification performance on several relevant settings where state of the art approaches fail.

| name | minimum | harmonic mean | geometric mean | arithmetic mean | maximum |
|---|---|---|---|---|---|
| $p$ | $p \to -\infty$ | $p = -1$ | $p \to 0$ | $p = 1$ | $p \to \infty$ |
| $m_p(a,b)$ | $\min\{a,b\}$ | $2\left(\frac{1}{a} + \frac{1}{b}\right)^{-1}$ | $\sqrt{ab}$ | $(a+b)/2$ | $\max\{a,b\}$ |

Table 1: Particular cases of scalar power means

Moreover, our approach scales to large datasets: even though the computation of the power mean Laplacian is in general prohibitive for large graphs, we present a matrix-free numerical scheme based on integral quadratures methods and Krylov subspace solvers which allows us to apply the power mean Laplacian regularizer to large sparse graphs. Finally, we perform numerical experiments on real world datasets and verify that our approach is competitive to state of the art approaches.

## 2  The Power Mean Laplacian

In this section we introduce our multilayer graph regularizer based on the power mean Laplacian. We define a multilayer graph $\mathbb{G}$ with $T$ layers as the set $\mathbb{G} = \{G^{(1)}, \ldots, G^{(T)}\}$, with each graph layer defined as $G^{(t)} = (V, W^{(t)})$, where $V = \{v_1, \ldots, v_n\}$ is the node set and $W^{(t)} \in \mathbb{R}_+^{n \times n}$ is the corresponding adjacency matrix, which we assume symmetric and nonnegative. We further denote the layers' normalized Laplacians as $L_{\text{sym}}^{(t)} = I - (D^{(t)})^{-1/2} W (D^{(t)})^{-1/2}$, where $D^{(t)}$ is the degree diagonal matrix with $(D^{(t)})_{ii} = \sum_{j=1}^{n} W_{ij}^{(t)}$.

The scalar power mean is a one-parameter family of scalar means defined as

$$m_p(x_1, \ldots, x_T) = \left(\tfrac{1}{T} \sum_{i=1}^{T} x_i^p\right)^{1/p}$$

where $x_1, \ldots, x_T$ are nonnegative scalars and $p$ is a real parameter. Particular choices of $p$ yield specific means such as the arithmetic, geometric and harmonic means, as illustrated in Table 1.

The **Power Mean Laplacian**, introduced in [19], is a matrix extension of the scalar power mean applied to the Laplacians of a multilayer graph and proposed as a more robust way to blend the information encoded across the layers. It is defined as

$$L_p = \left(\tfrac{1}{T} \sum_{i=1}^{T} (L_{\text{sym}}^{(i)})^p\right)^{1/p}$$

where $A^{1/p}$ is the unique positive definite solution of the matrix equation $X^p = A$. For the case $p \le 0$ a small diagonal shift $\varepsilon > 0$ is added to each Laplacian, i.e. we replace $L_{\text{sym}}^{(i)}$ with $L_{\text{sym}}^{(i)} + \varepsilon$, to ensure that $L_p$ is well defined as suggested in [3]. In what follows all the proofs hold for an arbitrary shift. Following [19], we set $\varepsilon = \log_{10}(1 + |p|) + 10^{-6}$ for $p \le 0$ in the numerical experiments.

## 3  Multilayer Semi-Supervised Learning with the Power Mean Laplacian

In this paper we consider the following optimization problem for the task of semi-supervised learning in multilayer graphs: Given $k$ classes $r = 1, \ldots, k$ and membership vectors $Y^{(r)} \in \mathbb{R}^n$ defined by $Y_i^{(r)} = 1$ if node $v_i$ belongs to class $r$ and $Y_i^{(r)} = 0$ otherwise, we let

$$f^{(r)} = \arg\min_{f \in \mathbb{R}^n} \|f - Y^{(r)}\|^2 + \lambda f^T L_p f. \tag{1}$$

The final class assignment for an unlabeled node $v_i$ is $y_i = \arg\max\{f_i^{(1)}, \ldots, f_i^{(k)}\}$. Note that the solution $f$ of (1), for a particular class $r$, is such that $(I + \lambda L_p)f = Y^{(r)}$. Equation (1) has two terms: the first term is a loss function based on the labeled nodes whereas the second term is a regularization term based on the power mean Laplacian $L_p$, which accounts for the multilayer graph structure. It is worth noting that the Local-Global approach of [32] is a particular case of our approach when only one layer ($T = 1$) is considered. Moreover, not that when $p = 1$ we obtain a regularizer term based on the arithmetic mean of Laplacians $L_1 = \frac{1}{T} \sum_{i=1}^{T} L_{\text{sym}}^{(i)}$. In the following section we analyze our proposed approach (1) under the Multilayer Stochastic Block Model.

# 4 Multilayer Stochastic Block Model

In this section we provide an analysis of semi-supervised learning for multilayer graphs with the power mean Laplacian as a regularizer under the Multilayer Stochastic Block Model (**MSBM**). The MSBM is a generative model for graphs showing certain prescribed clusters/classes structures via a set of membership parameters $p_{\text{in}}^{(t)}$ and $p_{\text{out}}^{(t)}$, $t = 1, \ldots, T$. These parameters designate the edge probabilities: given nodes $v_i$ and $v_j$ the probability of observing an edge between them on layer $t$ is $p_{\text{in}}^{(t)}$ (resp. $p_{\text{out}}^{(t)}$), if $v_i$ and $v_j$ belong to the same (resp. different) cluster/class. Note that, unlike the Labeled Stochastic Block Model [11], the MSBM allows multiple edges between the same pairs of nodes across the layers. For SSL with one layer under the SBM we refer the reader to [12, 22, 26].

We present an analysis in expectation. We consider $k$ clusters/classes $\mathcal{C}_1, \ldots, \mathcal{C}_k$ of equal size $|\mathcal{C}| = n/k$. We denote with calligraphic letters the layers of a multilayer graph in expectation $E(\mathbb{G}) = \{E(G^{(1)}, \ldots, E(G^{(T)})\}$, i.e. $\mathcal{W}^{(t)}$ is the expected adjacency matrix of the $t^{th}$-layer. We assume that our multilayer graphs are non-weighted, i.e. edges are zero or one, and hence we have $\mathcal{W}_{ij}^{(t)} = p_{\text{in}}^{(t)}$, (resp. $\mathcal{W}_{ij}^{(t)} = p_{\text{out}}^{(t)}$) for nodes $v_i, v_j$ belonging to the same (resp. different) cluster/class.

In order to grasp how different methods classify the nodes in multilayer graphs following the MSBM we analyze two different settings. In the first setting (Section 4.1) all layers have the same class structure and we study the conditions for different regularizers $L_p$ to correctly predict class labels. We further show that our approach is robust against the presence of noise layers, in the sense that it achieves a small classification error when at least one layer is informative and the remaining layers are potentially just noise. In this setting we distinguish the case where each class has the same amount of initial labels and the case where different classes have different number of labels. In the second setting (Section 4.2) we consider the case where each layer taken alone would lead to a large classification error whereas considering all the layers together can lead to a small classification error.

## 4.1 Complementary Information Layers

A common assumption in multilayer semi-supervised learning is that at least one layer encodes relevant information in the label prediction task. The next theorem discusses the classification error of the expected power mean Laplacian regularizer in this setting.

**Theorem 1.** *Let $E(\mathbb{G})$ be the expected multilayer graph with $T$ layers following the multilayer SBM with $k$ classes $\mathcal{C}_1, \ldots, \mathcal{C}_k$ of equal size and parameters $\left(p_{\text{in}}^{(t)}, p_{\text{out}}^{(t)}\right)_{t=1}^{T}$. Assume the same number of labeled nodes are available per class. Then, the solution of* (1) *yields zero test error if and only if*

$$m_p(\boldsymbol{\rho_\epsilon}) < 1 + \epsilon, \tag{2}$$

*where $(\boldsymbol{\rho_\epsilon})_t = 1 - (p_{\text{in}}^{(t)} - p_{\text{out}}^{(t)})/(p_{\text{in}}^{(t)} + (k-1)p_{\text{out}}^{(t)}) + \epsilon$, and $t = 1, \ldots, T$.*

This theorem shows that the power mean Laplacian regularizer allows to correctly classify the nodes if $p$ is such that condition (2) holds. In order to better understand how this condition changes when $p$ varies, we analyze in the next corollary the limit cases $p \to \pm\infty$.

**Corollary 1.** *Let $E(\mathbb{G})$ be an expected multilayer graph as in Theorem 1. Then,*

- *For $p \to \infty$, the test error is zero if and only if $p_{\text{out}}^{(t)} < p_{\text{in}}^{(t)}$ for all $t = 1, \ldots, T$.*

- *For $p \to -\infty$, the test error is zero if and only there exists a $t \in \{1, \ldots, T\}$ such that $p_{\text{out}}^{(t)} < p_{\text{in}}^{(t)}$.*

This corollary implies that the limit case $p \to \infty$ requires that *all layers* convey information regarding the clustering/class structure of the multilayer graph, whereas the case $p \to -\infty$ requires that *at least one layer* encodes clustering/class information, and hence it is clear that conditions for the limit $p \to -\infty$ are less restrictive than the conditions for the limit case $p \to \infty$. The next Corollary shows that the smaller the power parameter $p$ is, the less restrictive are the conditions to yield a zero test error.

**Corollary 2.** *Let $E(\mathbb{G})$ be an expected multilayer graph as in Theorem 1. Let $p \leq q$. If $\mathcal{L}_q$ yields zero test error, then $\mathcal{L}_p$ yields a zero test error.*

The previous results show the effectivity of the power mean Laplacian regularizer in expectation. We now present a numerical evaluation based on Theorem 1 and Corollaries 1 and 2 on random

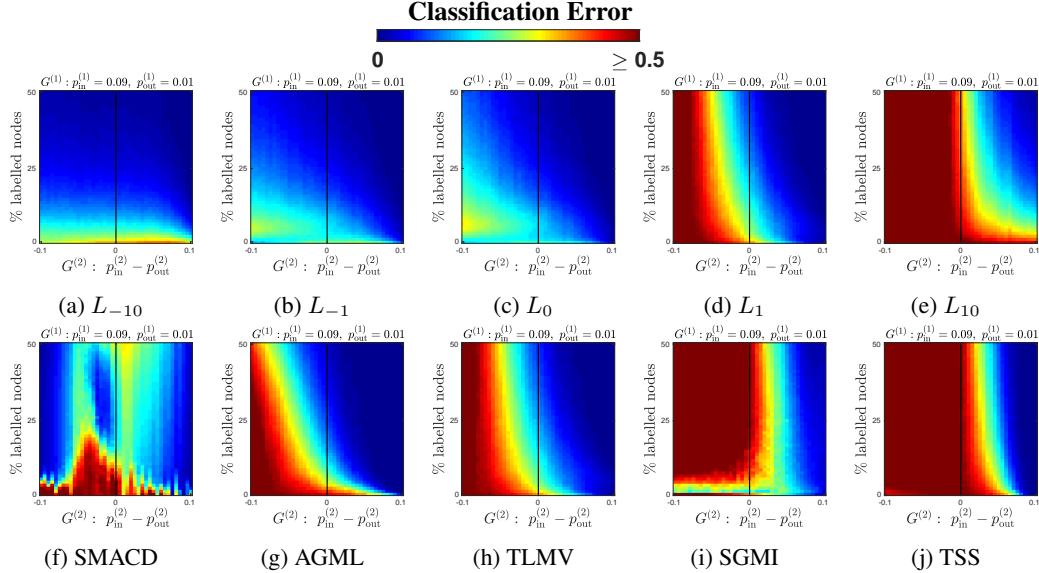

Figure 1: Average classification error under the Stochastic Block Model computed from 100 runs. **Top Row:** Particular cases with the power mean Laplacian. **Bottom Row:** State of the art models.

graphs sampled from the SBM. The corresponding results are presented in Fig. 1 for classification with regularizers $L_{-10}, L_{-1}, L_0, L_1, L_{10}$ and $\lambda = 1$. We first describe the setting we consider: we generate random multilayer graphs with two layers ($T = 2$) and two classes ($k = 2$) each composed by 100 nodes ($|\mathcal{C}| = 100$). For each parameter configuration ($p_{\text{in}}^{(1)}, p_{\text{out}}^{(1)}, p_{\text{in}}^{(2)}, p_{\text{out}}^{(2)}$) we generate 10 random multilayer graphs and 10 random samples of labeled nodes, yielding a total of 100 runs per parameter configuration, and report the average test error. Our goal is to evaluate the classification performance under different SBM parameters and different amounts of labeled nodes. To this end, we fix the first layer $G^{(1)}$ to be informative of the class structure ($p_{\text{in}}^{(1)} - p_{\text{out}}^{(1)} = 0.08$), i.e. one can achieve a low classification error by taking this layer alone, provided sufficiently many labeled nodes are given. The second layer will go from non-informative (noisy) configurations ($p_{\text{in}}^{(2)} < p_{\text{out}}^{(2)}$, left half of $x$-axis) to informative configurations ($p_{\text{in}}^{(2)} > p_{\text{out}}^{(2)}$, right half of $x$-axis), with $p_{\text{in}}^{(t)} + p_{\text{out}}^{(t)} = 0.1$ for both layers. Moreover, we consider different amounts of labeled nodes: going from 1% to 50% ($y$-axis). The corresponding results are presented in Figs. 1a,1b,1c,1d, and 1e.

In general one can expect a low classification error when both layers $G^{(1)}$ and $G^{(2)}$ are informative (right half of $x$-axis). We can see that this is the case for all power mean Laplacian regularizers here considered (see top row of Fig. 1). In particular, we can see in Fig. 1e that $L_{10}$ performs well only when **both** layers are informative and completely fails when the second layer is not informative, regardless of the amount of labeled nodes. On the other side we can see in Fig. 1a that $L_{-10}$ achieves in general a low classification error, regardless of the configuration of the second layer $G^{(2)}$, i.e. when $G^{(1)}$ **or** $G^{(2)}$ are informative. Moreover, we can see that overall the areas with low classification error (dark blue) increase when the parameter $p$ decreases, verifying the result from Corollary 2. In the bottom row of Fig. 1 we present the performance of state of the art methods. We can observe that most of them present a classification performance that resembles the one of the power mean Laplacian regularizer $L_1$. In general their classification performance drops when the level of noise increases, i.e. for non-informative configurations of the second layer $G^{(2)}$, and they are outperformed by the power mean Laplacian regularizer for small values of $p$.

**Unbalanced Class Proportion on Labeled Datasets.** In the previous analysis we assumed that we had the same amount of labeled nodes per class. We consider now the case where the number of labeled nodes per class is different. This setting was considered in [35], where the goal was to overcome unbalanced class proportions in labeled nodes. To this end, they propose a Class Mass Normalization (CMN) strategy, whose performance was also tested in [34]. In the following result we show that, provided the ground truth classes have the same size, different amounts of labeled nodes per class affect the conditions in expectation for zero classification error of (1). For simplicity, we consider here only the case of two classes.

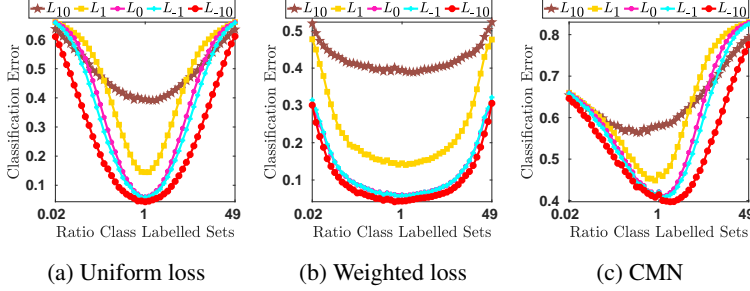

| (a) Uniform loss | (b) Weighted loss | (c) CMN |

**Figure 2:** Different class weighted loss strategies. Left to right: uniform loss, weighted loss, and Class Mass Normalization.

**Theorem 2.** *Let $E(\mathbb{G})$ be the expected multilayer graph with $T$ layers following the multilayer SBM with two classes $\mathcal{C}_1, \mathcal{C}_2$ of equal size and parameters $\left(p_{\text{in}}^{(t)}, p_{\text{out}}^{(t)}\right)_{t=1}^{T}$. Assume $n_1, n_2$ nodes from $\mathcal{C}_1, \mathcal{C}_2$ are labeled, respectively. Let $\lambda = 1$. Then (1) yields zero test error if*

$$m_p(\boldsymbol{\rho_\epsilon}) < \min\left\{\frac{n_1}{n_2}, \frac{n_2}{n_1}\right\} \tag{3}$$

*where $(\boldsymbol{\rho_\epsilon})_t = 1 - (p_{\text{in}}^{(t)} - p_{\text{out}}^{(t)})/(p_{\text{in}}^{(t)} + (k-1)p_{\text{out}}^{(t)}) + \epsilon$, and $t = 1, \ldots, T$.*

Observe that Theorem 2 provides only a sufficient condition. A necessary and sufficient condition for zero test error in terms of $p$, $n_1$ and $n_2$ is given in the supplementary material.

A different objective function can be employed for the case of classes with different number of labels per class. Let $C$ be the diagonal matrix defined by $C_{ii} = n/n_r$, if node $v_i$ has been labeled to belong to class $\mathcal{C}_r$. Consider the following modification of (1)

$$\arg\min_{f \in \mathbb{R}^n} \|f - CY\|^2 + \lambda f^T L_p f \tag{4}$$

The next Theorem shows that using (4) in place of (1) allows us to retrieve the same condition of Theorem 1 for zero test error in expectation in the setting where the number of labeled nodes per class are not equal.

**Theorem 3.** *Let $E(\mathbb{G})$ be the expected multilayer graph with $T$ layers following the multilayer SBM $k$ classes $\mathcal{C}_1, \ldots, \mathcal{C}_k$ of equal size and parameters $\left(p_{\text{in}}^{(t)}, p_{\text{out}}^{(t)}\right)_{t=1}^{T}$. Let $n_1, \ldots, n_k$ be the number of labeled nodes per class. Let $C \in \mathbb{R}^{n \times n}$ be a diagonal matrix with $C_{ii} = n/n_r$ for $v_i \in \mathcal{C}_r$. The solution to (4) yields a zero test classification error if and only if*

$$m_p(\boldsymbol{\rho_\epsilon}) < 1 + \epsilon, \tag{5}$$

*where $(\boldsymbol{\rho_\epsilon})_t = 1 - (p_{\text{in}}^{(t)} - p_{\text{out}}^{(t)})/(p_{\text{in}}^{(t)} + (k-1)p_{\text{out}}^{(t)}) + \epsilon$, and $t = 1, \ldots, T$.*

In Figs. 2a, 2b, and 2c. we present a numerical experiment with random graphs of our analysis in expectation. We consider the following setting: we generate multilayer graphs with two layers ($T = 2$) and two classes ($k = 2$) each composed by 100 nodes ($|\mathcal{C}| = 100$). We fix $p_{\text{in}}^{(1)} - p_{\text{out}}^{(1)} = 0.08$ and $p_{\text{in}}^{(2)} - p_{\text{out}}^{(2)} = 0$, with $p_{\text{in}}^{(t)} + p_{\text{out}}^{(t)} = 0.1$ for both layers. We fix the total amount of labeled nodes to be $n_1 + n_2 = 50$ and let $n_1, n_2 = 1, \ldots 49$. For each setting we generate 10 multilayer graphs and 10 sets of labeled nodes, yielding a total of 100 runs per setting, and report the average test classification error. In Fig. 2a we can see the performance of the power mean Laplacian regularizer without modifications. We can observe how different proportions of labeled nodes per class affect the performance. In Fig. 2b, we present the performance of the modified approach (4) and observe that it yields a better performance against different class label proportions. Finally in Fig. 2c we present the performance based on Class Mass Normalization [1], where we can see that its effect is slightly skewed to one class and its overall performance is larger than the proposed approach.

## 4.2 Information-Independent Layers

In the previous section we considered the case where at least one layer had enough information to correctly estimate node class labels. In this section we now consider the case where single layers

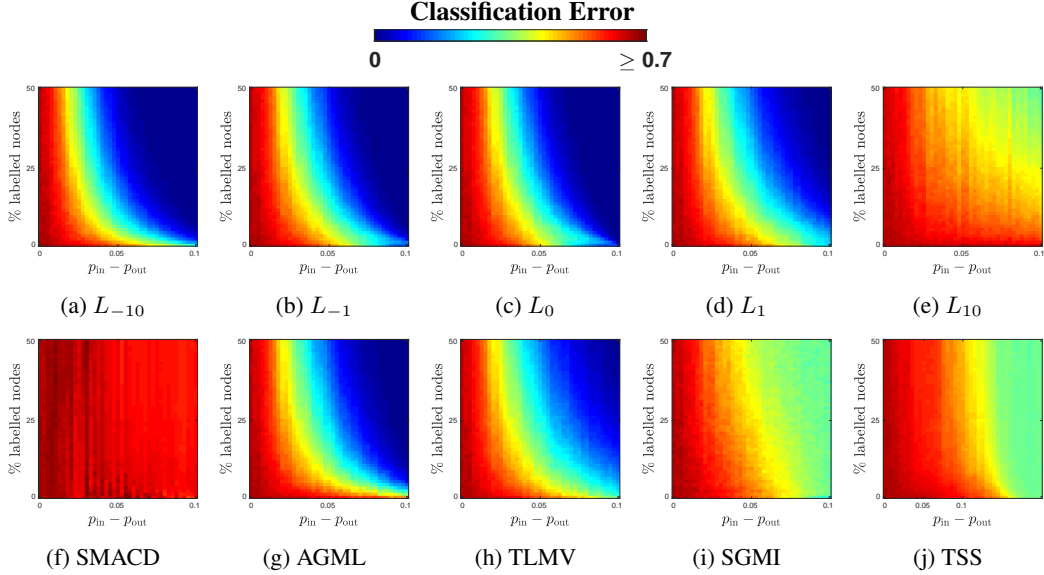

Figure 4: Average test error under the SBM.Multilayer graph with 3 layers and 3 classes.**Top Row:** Particular cases with the power mean Laplacian. **Bottom Row:** State of the art models.

taken alone obtain a large classification error, whereas when all the layers are taken together it is possible to obtain a good classification performance. For this setting we consider multilayer graphs with 3 layers ($T = 3$) and three classes ($k = 3$) $\mathcal{C}_1, \mathcal{C}_2, \mathcal{C}_3$, each composed by 100 nodes ($|\mathcal{C}| = 100$) with the following expected adjacency matrix per layer:

$$\mathcal{W}_{i,j}^{(t)} = \begin{cases} p_{\text{in}}, & v_i, v_j \in \mathcal{C}_t \text{ or } v_i, v_j \in \overline{\mathcal{C}_t} \\ p_{\text{out}}, & \text{else} \end{cases} \tag{6}$$

for $t = 1, 2, 3$, i.e. layer $G^{(t)}$ is informative of class $\mathcal{C}_t$ but not of the remaining classes, and hence any classification method using one single layer will provide a poor classification performance. In Fig. 4 we present numerical experiments: for each parameter setting $(p_{\text{in}}, p_{\text{out}})$ we generate 5 multilayer graphs together with 5 samples of labeled nodes yielding a total of 25 runs per setting, and report the average test classification error. Also in this case we observe that the power mean Laplacian regularizer does identify the global class structure and that it leverages the information provided by labeled nodes, particularly for smaller values of $p$. On the other hand, this is not the case for all other state of the art methods. In fact, we can see that SGMI and TSS performs similarly to $L_{10}$ which has the largest classification error. Moreover, we can see that AGML and TLMV perform similarly to the arithmetic mean of Laplacians $L_1$, which in turn is outperformed by the power mean Laplacian regularizer $L_{-10}$. Please see the supplementary material for a more detailed comparison.

## 5    A Scalable Matrix-free Numerical Method for the System $(I + \lambda L_p)f = Y$

In this section we introduce a matrix-free method for the solution of the system $(I + \lambda L_p)f = Y$ based on contour integrals and Krylov subspace methods. The method exploits the sparsity of the Laplacians of each layer and is matrix-free, in the sense that it requires only to compute the matrix-vector product $L_{\text{sym}}^{(i)} \times vector$, without requiring to store the matrices. Thus, when the layers are sparse, the method scales to large datasets. Observe that this is a critical requirement as $L_p$ is in general a dense matrix, even for very sparse layers, and thus computing and storing $L_p$ is very prohibitive for large multilayer graphs. We present a method for negative integer values $p < 0$, leaving aside the limit case $p \rightarrow 0$ as it requires a particular treatment. The following is a brief overview of the proposed approach. Further details are available in the supplementary material.

Let $A_1, \ldots, A_T$ be symmetric positive definite matrices, $\varphi : \mathbb{C} \rightarrow \mathbb{C}$ defined by $\varphi(z) = z^{1/p}$ and $L_p = T^{-1/p}\varphi(S_p)$, where $S_p = A_1^p + \cdots + A_T^p$. The proposed method consists of three main steps:

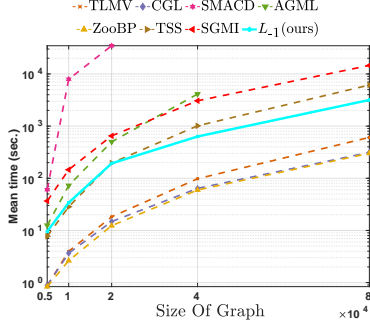

Figure 5: Mean execution time of 10 runs for different methods. $L_{-1}$(ours) stands for the power mean Laplacian regularizer together with our proposed matrix-free contour integral based method. We generate multilayer graphs with two layers, each with two classes of same size with parameters $p_{\text{in}} = 0.05$ and $p_{\text{in}} = 0.025$ and graphs of of sizes $[0.5, 1, 2, 4, 8] \times 10^4$. Observe that our matrix free approach for $L_{-1}$ (solid blue curve) is competitive to state of the art approaches as TSS[28], outperforming AGML[23], SGMI[13] and SMACD[9]. For TLMV[33] and SGMI we use our own implementation.

1. We solve the system $(I + \lambda L_p)^{-1} Y$ via a Krylov method (e.g. PCG or GMRES) with convergence rate $O((\frac{\kappa^2 - 1}{\kappa^2})^{h/2})$ [25], where $\kappa = \lambda_{\max}(L_p)/\lambda_{\min}(L_p)$. At iteration $h$, this method projects the problem onto the Krylov subspace spanned by $\{Y, \lambda L_p Y, (\lambda L_p)^2 Y, \ldots, (\lambda L_p)^h Y\}$, and efficiently solve the projected problem.

2. The previous step requires the matrix-vector product $L_p Y = T^{-1/p} \varphi(S_p) Y$ which we compute by approximating the Cauchy integral form of the function $\varphi$ with the trapezoidal rule in the complex plane [10]. Taking $N$ suitable contour points and coefficients $\beta_0, \ldots, \beta_N$, we have

$$\varphi_N(S_p)Y = \beta_0 S_p \, \text{Im} \left\{ \sum_{i=1}^{N} \beta_i (z_i^2 I - S_p)^{-1} Y \right\}, \tag{7}$$

which has geometric convergence [10]: $\|\varphi(S_p)Y - \varphi_N(S_p)Y\| = O(e^{-2\pi^2 N/(\ln(M/m)+6)})$, where $m, M$ are such that $M \geq \lambda_{\max}(S_p)$ and $m \leq \lambda_{\min}(S_p)$.

3. The previous step requires to solve linear systems of the form $(zI - S_p)^{-1} Y$. We solve each of these systems via a Krylov subspace method, projecting, at each iteration $h$, onto the subspace spanned by $\{Y, S_p Y, S_p^2 Y, \ldots, S_p^h Y\}$. Since $S_p = \sum_{i=1}^{T} A_i^{-|p|}$ this problem reduces to computing $|p|$ linear systems with $A_i$ as coefficient matrix, for $i = 1 \ldots, T$. Provided that $A_1, \ldots, A_T$ are sparse matrices, this is done efficiently using pcg with incomplete Cholesky preconditioners.

Notice that the method allows a high level of parallelism. In fact, the $N$ (resp. $p$) linear systems solvers at step 2 (resp. 3) are independent and can be run in parallel. Moreover, note that the main task of the method is solving linear systems with Laplacian matrices, which can be solved linearly in the number of edges in the corresponding adjacency matrix. Hence, the proposed approach scales to large sparse graphs and is highly parallelizable. A time execution analysis is provided in Fig 5, where we can see that the time execution of our approach is competitive to the state of the art as TSS[28], outperforming AGML[23], SGMI[13] and SMACD[9].

## 6 Experiments on Real Datasets

In this section we compare the performance of the proposed approach with state of the art methods on real world datasets. We consider the following datasets: *3-sources* [16], which consists of news articles that were covered by news sources BBC, Reuters and Guardian; *BBC*[7] and *BBC Sports*[8] news articles, a dataset of Wikipedia articles with ten different classes [24], the hand written *UCI* digits dataset with six different set of features, and citations datasets *CiteSeer*[17], *Cora*[18] and *WebKB*(Texas)[5]. For each dataset we build the corresponding layer adjacency matrices by taking the symmetric $k$-nearest neighbour graph using as similarity measure the Pearson linear correlation, (i.e. we take the $k$ neighbours with highest correlation), and take the unweighted version of it. Datasets CiteSeer, Cora and WebKB have only two layers, where the first one is a fixed precomputed citation layer, and the second one is the corresponding $k$-nearest neighbour graph built from document features.

As **baseline methods** we consider: TSS [28] which identifies an optimal linear combination of graph Laplacians, SGMI [13] which performs label propagation by sparse integration, TLMV [33] which is a weighted arithmetic mean of adjacency matrices, CGL [1] which is a convex combination of the pseudo inverse Laplacian kernel, AGML [23] which is a parameter-free method for optimal graph layer weights, ZooBP [6] which is a fast approximation of Belief Propagation, and SMACD [9] which is a tensor factorization method designed for semi-supervised learning. Finally we set parameters for TSS to ($c = 10, c_0 = 0.4$), SMACD ($\lambda = 0.01$)[2], TLMV ($\lambda = 1$), SGMI ($\lambda_1 = 1, \lambda_2 = 10^{-3}$)

| 3sources | 1% | 5% | 10% | 15% | 20% | 25% |
|---|---|---|---|---|---|---|
| TLMV | 29.8 | 21.5 | **20.8** | 20.3 | 15.5 | 16.5 |
| CGL | 50.2 | 45.5 | 36.4 | 30.6 | 23.8 | 19.8 |
| SMACD | 91.5 | 91.1 | 91.2 | 90.9 | 90.7 | 91.3 |
| AGML | **23.9** | 26.3 | 33.9 | 33.3 | 26.1 | 22.0 |
| ZooBP | 31.0 | 21.9 | 21.3 | 19.8 | 15.0 | 15.3 |
| TSS | 29.8 | 23.9 | 33.1 | 34.6 | 34.8 | 35.0 |
| SGMI | 34.4 | 26.6 | 25.4 | 24.4 | 19.1 | 17.9 |
| $L_1$ | 33.5 | 23.9 | 23.4 | 20.1 | 15.6 | **14.6** |
| $L_{-1}$ | 28.4 | **20.0** | 21.8 | 22.0 | 17.2 | 17.9 |
| $L_{-10}$ | 40.9 | 29.1 | 21.9 | **19.3** | **14.8** | 14.7 |

| BBC | 1% | 5% | 10% | 15% | 20% | 25% |
|---|---|---|---|---|---|---|
| TLMV | **29.0** | 19.3 | 13.2 | 11.1 | 9.3 | 8.8 |
| CGL | 72.5 | 52.3 | 36.1 | 27.4 | 22.0 | 17.1 |
| SMACD | 74.4 | 73.5 | 72.8 | 72.6 | 72.5 | 72.4 |
| AGML | 60.0 | 34.2 | 18.6 | 13.1 | 11.0 | 9.5 |
| ZooBP | 31.1 | 20.1 | 15.0 | 12.2 | 10.0 | 9.1 |
| TSS | 40.4 | 26.1 | 20.9 | 20.1 | 19.8 | 19.7 |
| SGMI | 37.6 | 28.9 | 24.9 | 22.8 | 20.7 | 19.3 |
| $L_1$ | 31.3 | 22.8 | 17.4 | 13.5 | 10.2 | 8.9 |
| $L_{-1}$ | 31.0 | **17.0** | **11.5** | **10.5** | **9.2** | **8.7** |
| $L_{-10}$ | 51.6 | 26.9 | 16.6 | 12.8 | 10.3 | 9.5 |

| BBCS | 1% | 5% | 10% | 15% | 20% | 25% |
|---|---|---|---|---|---|---|
| TLMV | 25.6 | 12.6 | 10.5 | 7.5 | 6.4 | 5.4 |
| CGL | 79.2 | 51.6 | 34.9 | 23.4 | 16.5 | 12.7 |
| SMACD | 77.8 | 80.6 | 82.4 | 96.4 | 98.4 | 98.3 |
| AGML | 34.6 | 17.4 | 12.1 | 7.0 | 6.0 | 5.4 |
| ZooBP | 33.8 | 13.9 | 11.3 | 8.8 | 7.6 | 6.2 |
| TSS | 23.9 | 13.2 | 14.1 | 12.3 | 13.1 | 12.2 |
| SGMI | 31.9 | 19.6 | 16.6 | 15.5 | 14.8 | 12.1 |
| $L_1$ | 29.9 | 15.0 | 13.5 | 10.6 | 8.7 | 7.2 |
| $L_{-1}$ | 23.8 | 11.6 | **8.7** | **6.3** | **5.8** | **5.1** |
| $L_{-10}$ | 48.7 | 22.5 | 14.2 | 9.1 | 7.8 | 6.1 |

| Wikipedia | 1% | 5% | 10% | 15% | 20% | 25% |
|---|---|---|---|---|---|---|
| TLMV | 65.7 | 56.8 | 46.4 | 43.1 | 40.8 | 39.2 |
| CGL | 87.3 | 83.0 | 82.5 | 82.2 | 83.0 | 83.0 |
| SMACD | 85.4 | 85.6 | 85.4 | 85.3 | 86.8 | 90.0 |
| AGML | 71.3 | 66.6 | 48.1 | 42.1 | 38.4 | 37.3 |
| ZooBP | 67.6 | 58.0 | 47.0 | 43.8 | 41.2 | 39.8 |
| TSS | 87.7 | 84.7 | 83.3 | 81.9 | 82.3 | 81.4 |
| SGMI | 69.3 | 84.8 | 84.5 | 83.8 | 83.2 | 82.8 |
| $L_1$ | 68.2 | 61.1 | 53.6 | 48.3 | 44.1 | 42.3 |
| $L_{-1}$ | **59.1** | **52.3** | **40.2** | **36.3** | **35.1** | **34.1** |
| $L_{-10}$ | 66.9 | 57.2 | 43.2 | 38.7 | 36.3 | 34.9 |

| UCI | 1% | 5% | 10% | 15% | 20% | 25% |
|---|---|---|---|---|---|---|
| TLMV | 28.9 | 20.4 | 16.3 | 14.4 | 13.7 | 12.7 |
| CGL | 81.8 | 64.0 | 54.6 | 49.1 | 46.7 | 46.7 |
| SMACD | 73.6 | 81.0 | 90.0 | 90.0 | 86.2 | 81.9 |
| AGML | 25.3 | 17.2 | 15.2 | 13.2 | 12.5 | 12.0 |
| ZooBP | 30.8 | 21.7 | 17.6 | 15.1 | 14.1 | 13.0 |
| TSS | 24.0 | 17.6 | 16.6 | 15.9 | 15.8 | 15.6 |
| SGMI | 36.0 | 44.4 | 50.9 | 50.4 | 50.2 | 48.8 |
| $L_1$ | 31.3 | 23.8 | 18.7 | 15.6 | 14.4 | 13.2 |
| $L_{-1}$ | 30.5 | **17.1** | **13.8** | **12.6** | **12.3** | **11.9** |
| $L_{-10}$ | 57.0 | 33.8 | 23.7 | 17.6 | 15.3 | 13.4 |

| Citeseer | 1% | 5% | 10% | 15% | 20% | 25% |
|---|---|---|---|---|---|---|
| TLMV | 51.5 | 39.4 | 36.5 | 33.7 | 31.6 | 30.3 |
| CGL | 89.3 | 64.7 | 58.0 | 49.8 | 44.5 | 40.9 |
| SMACD | 90.7 | 90.4 | 67.0 | 65.5 | 66.8 | 68.9 |
| AGML | **47.3** | **32.3** | **29.6** | **28.2** | **27.5** | **27.0** |
| ZooBP | 63.6 | 41.9 | 38.7 | 35.8 | 33.8 | 32.2 |
| TSS | 58.5 | 49.5 | 45.9 | 42.1 | 39.8 | 38.4 |
| SGMI | 59.4 | 46.8 | 44.0 | 42.3 | 40.5 | 39.2 |
| $L_1$ | 56.3 | 44.1 | 41.2 | 38.5 | 36.1 | 34.7 |
| $L_{-1}$ | 52.4 | 39.0 | 35.6 | 32.6 | 30.9 | 29.5 |
| $L_{-10}$ | 68.6 | 54.6 | 48.5 | 43.0 | 39.7 | 37.2 |

| Cora | 1% | 5% | 10% | 15% | 20% | 25% |
|---|---|---|---|---|---|---|
| TLMV | 46.0 | 34.1 | 28.8 | 25.8 | 22.5 | 20.6 |
| CGL | 85.5 | 70.1 | 56.5 | 49.1 | 44.2 | 40.0 |
| SMACD | 75.6 | 76.7 | 78.7 | 78.7 | 81.0 | 87.1 |
| AGML | 54.7 | 36.0 | 25.4 | **20.7** | **18.1** | **16.5** |
| ZooBP | 54.7 | 38.0 | 32.9 | 30.2 | 27.6 | 26.2 |
| TSS | **38.8** | **27.7** | **24.1** | 21.5 | 20.0 | 19.1 |
| SGMI | 57.3 | 47.7 | 43.0 | 41.8 | 40.1 | 38.5 |
| $L_1$ | 50.7 | 38.2 | 33.4 | 31.2 | 28.2 | 25.6 |
| $L_{-1}$ | 43.2 | 31.8 | 24.5 | 21.1 | 18.8 | 17.2 |
| $L_{-10}$ | 62.0 | 46.3 | 35.4 | 29.4 | 25.2 | 22.3 |

| WebKB | 1% | 5% | 10% | 15% | 20% | 25% |
|---|---|---|---|---|---|---|
| TLMV | 58.6 | 49.4 | 45.6 | 47.2 | 47.6 | 48.2 |
| CGL | 80.4 | 82.4 | 84.4 | 86.9 | 82.7 | 89.2 |
| SMACD | 87.3 | 87.2 | 87.2 | 87.4 | 87.8 | 87.8 |
| AGML | 56.5 | 50.3 | 46.8 | 44.7 | 47.6 | 46.8 |
| ZooBP | 52.0 | 45.0 | 38.7 | 38.5 | **36.4** | **33.5** |
| TSS | 60.9 | 51.0 | 50.5 | 47.3 | 49.2 | 48.7 |
| SGMI | **44.9** | **39.7** | 41.9 | **34.9** | 40.3 | 52.5 |
| $L_1$ | 58.5 | 49.0 | 44.8 | 44.3 | 44.5 | 44.4 |
| $L_{-1}$ | 49.9 | 45.5 | 40.7 | 39.5 | 39.9 | 40.3 |
| $L_{-10}$ | 52.3 | 41.9 | **38.0** | 38.1 | 36.8 | 39.5 |

Table 2: Experiments in real datasets. Notation: **best** performances are marked with bold fonts and gray background and second best performances with only gray background.

and $\lambda = 0.1$ for $L_1$ and $\lambda = 10$ for $L_{-1}$ and $L_{-10}$. We do not perform cross validation in our experimental setting due to the large execution time in some of the methods here considered. Hence we fix the parameters for each method in all experiments.

We fix nearest neighbourhood size to $k = 10$ and generate 10 samples of labeled nodes, where the percentage of labeled nodes per class is in the range $\{1\%, 5\%, 10\%, 15\%, 20\%, 25\%\}$. The average test errors are presented in table 2, where the **best** (resp. second best) performances are marked with bold fonts and gray background (resp. with only gray background). We can see that the first and second best positions are in general taken by the power mean Laplacian regularizers $L_1, L_{-1}, L_{-10}$, being clear for all datasets except with 3-sources. Moreover we can see that in $77\%$ of all cases $L_{-1}$ presents either the best or the second best performance, further verifying that our proposed approach based on the power mean Laplacian for semi-supervised learning in multilayer graph is a competitive alternative to state of the art methods[3].

**Acknowledgement** P.M and M.H are supported by the DFG Cluster of Excellence "Machine Learning – New Perspectives for Science", EXC 2064/1, project number 390727645

## Footnotes

[1]We follow the authors' implementation: http://pages.cs.wisc.edu/~jerryzhu/pub/harmonic_function.m

[2]this is the default value in the code released by the authors: https://github.com/egujr001/SMACD

[3]Communications with the authors of [9] could not clarify the bad performance of SMACD.

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
