[Supplementary Material]

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

# Appendices

This section contains all the proofs of results mentioned in the main paper, together with a detailed description of the proposed numerical scheme. It is organized as follows:

- Section A contains auxiliary results,
- Section B contains the proof of Theorem 1,
- Section C contains the proof of Corollary 1,
- Section D contains the proof of Corollary 2,
- Section E contains general version of Theorem 2
- Section F contains the proof of Theorem 2,
- Section G contains the proof of Theorem 3.
- Section H contains a detailed exposition of the numerical scheme presented in Section 5.
- Section I we present a numerical analysis for the case of three layers and three classes as in Sec. 4.2
- Section J presents a numerical analysis of the effect of the regularization parameter $\lambda$

## A   Auxiliary Results

We first present some results that will be useful.

The following theorem states the monotonicity of the scalar power mean.

**Theorem 4** ([4], Ch. 3, Thm. 1). *Let $p < q$ then $m_p(a, b) \leq m_q(a, b)$ with equality if and only if $a = b$.*

The following lemma shows the effect of the matrix power mean when matrices have a common eigenvector.

**Lemma 1** ([21]). *Let $\mathbf{u}$ be an eigenvector of $A_1, \ldots, A_T$ with corresponding eigenvalues $\lambda_1, \ldots, \lambda_T$. Then $\mathbf{u}$ is an eigenvector of $M_p(A_1, \ldots, A_T)$ with eigenvalue $m_p(\lambda_1, \ldots, \lambda_T)$.*

The following Lemma states the eigenvalues and eigenvectors of expected adjacency matrices according to the Stochastic Block Model here considered.

**Lemma 2.** *Let $\mathcal{C}_1, \ldots, \mathcal{C}_k$ be clusters of equal size $|\mathcal{C}| = n/k$. Let $\mathcal{W} \in \mathbb{R}^{n \times n}$ be defined as*

$$\mathcal{W} = (p_{\text{in}} - p_{\text{out}}) \sum_{i=1}^{k} \mathbf{1}_{\mathcal{C}_i} \mathbf{1}_{\mathcal{C}_i}^T + p_{\text{out}} \mathbf{1}\mathbf{1}^T \tag{8}$$

*and let $\boldsymbol{\chi}_1, \ldots, \boldsymbol{\chi}_k \in \mathbb{R}^n$ be defined as*

$$\boldsymbol{\chi}_1 = \mathbf{1}, \qquad \boldsymbol{\chi}_r = \sum_{j=1}^{r} \mathbf{1}_{\mathcal{C}_j} - r\mathbf{1}_{\mathcal{C}_r} \tag{9}$$

*for $r = 2, \ldots, k$. Then, $\boldsymbol{\chi}_1, \ldots, \boldsymbol{\chi}_k$ are orthogonal eigenvectors of $\mathcal{W}$, with eigenvalues*

$$\lambda_1 = |\mathcal{C}| (p_{\text{in}} + (k-1)p_{\text{out}}), \qquad \lambda_r = |\mathcal{C}| (p_{\text{in}} - p_{\text{out}}) \tag{10}$$

*Proof.* Please note that from the definition that the matrix $\mathcal{W}$ is equal to $p_{\text{in}}$ in the block diagonal and $p_{\text{out}}$ elsewhere. We first consider the following matrix vector products that can be easily verified:

$$\mathcal{W}\mathbf{1} = |\mathcal{C}| (p_{\text{in}} + (k-1)p_{\text{out}})\mathbf{1} \tag{11}$$
$$\mathcal{W}\mathbf{1}_{\mathcal{C}_i} = |\mathcal{C}| (p_{\text{in}}\mathbf{1}_{\mathcal{C}_i} + p_{\text{out}}\mathbf{1}_{\overline{\mathcal{C}_i}}) \tag{12}$$

Moreover, we can see that

$$
\begin{aligned}
\mathcal{W}\left(\mathbf{1}_{\mathcal{C}_j} - \mathbf{1}_{\mathcal{C}_i}\right) &= |\mathcal{C}| \left(\left(p_{\text{in}}\mathbf{1}_{\mathcal{C}_j} + p_{\text{out}}\mathbf{1}_{\overline{\mathcal{C}_j}}\right) - \left(p_{\text{in}}\mathbf{1}_{\mathcal{C}_i} + p_{\text{out}}\mathbf{1}_{\overline{\mathcal{C}_i}}\right)\right) \\
&= |\mathcal{C}| \left(p_{\text{in}}\left(\mathbf{1}_{\mathcal{C}_j} - \mathbf{1}_{\mathcal{C}_i}\right) + p_{\text{out}}\left(\mathbf{1}_{\overline{\mathcal{C}_j}} - \mathbf{1}_{\overline{\mathcal{C}_i}}\right)\right) \\
&= |\mathcal{C}| \left(p_{\text{in}}\left(\mathbf{1}_{\mathcal{C}_j} - \mathbf{1}_{\mathcal{C}_i}\right) - p_{\text{out}}\left(\mathbf{1}_{\mathcal{C}_i} - \mathbf{1}_{\mathcal{C}_j}\right)\right) \\
&= |\mathcal{C}| (p_{\text{in}} - p_{\text{out}})\left(\mathbf{1}_{\mathcal{C}_j} - \mathbf{1}_{\mathcal{C}_i}\right)
\end{aligned}
$$

Now we show that $\chi_2, \ldots, \chi_k$ are eigenvectors of $\mathcal{W}$.

$$\mathcal{W}\chi_r = \mathcal{W}\left(\sum_{j=1}^{r} \mathbf{1}_{\mathcal{C}_j} - r\mathbf{1}_{\mathcal{C}_r}\right)$$

$$= \mathcal{W}\sum_{j=1}^{r}(\mathbf{1}_{\mathcal{C}_j} - \mathbf{1}_{\mathcal{C}_r})$$

$$= \sum_{j=1}^{r}\mathcal{W}\left(\mathbf{1}_{\mathcal{C}_j} - \mathbf{1}_{\mathcal{C}_r}\right)$$

$$= \sum_{j=1}^{r}|\mathcal{C}|\left(p_{\text{in}} - p_{\text{out}}\right)\left(\mathbf{1}_{\mathcal{C}_j} - \mathbf{1}_{\mathcal{C}_r}\right)$$

$$= |\mathcal{C}|\left(p_{\text{in}} - p_{\text{out}}\right)\sum_{j=1}^{r}\left(\mathbf{1}_{\mathcal{C}_j} - \mathbf{1}_{\mathcal{C}_r}\right)$$

$$= |\mathcal{C}|\left(p_{\text{in}} - p_{\text{out}}\right)\left(\sum_{j=1}^{r}\mathbf{1}_{\mathcal{C}_j} - r\mathbf{1}_{\mathcal{C}_r}\right)$$

$$= |\mathcal{C}|\left(p_{\text{in}} - p_{\text{out}}\right)\chi_r$$

$$= \lambda_r\chi_r$$

Furthermore, we can see that eigenvectors $\chi_2, \ldots, \chi_k$ are orthogonal. Let $2 \leq r < s \leq k$, then

$$\chi_r^T\chi_s = \left(\sum_{j_1=1}^{r}\mathbf{1}_{\mathcal{C}_{j_1}} - r\mathbf{1}_{\mathcal{C}_r}\right)^T\left(\sum_{j_2=1}^{s}\mathbf{1}_{\mathcal{C}_{j_2}} - s\mathbf{1}_{\mathcal{C}_s}\right)$$

$$= \sum_{j_1=1}^{r}\sum_{j_2=1}^{s}\mathbf{1}_{\mathcal{C}_{j_1}}^T\mathbf{1}_{\mathcal{C}_{j_2}} - s\sum_{j_1=1}^{r}\mathbf{1}_{\mathcal{C}_{j_1}}^T\mathbf{1}_{\mathcal{C}_s} - r\sum_{j_2=1}^{s}\mathbf{1}_{\mathcal{C}_{j_2}}^T\mathbf{1}_{\mathcal{C}_r} + rs\mathbf{1}_{\mathcal{C}_r}^T\mathbf{1}_{\mathcal{C}_s}$$

$$= \sum_{j_1=1}^{r}\sum_{j_2=1}^{s}\mathbf{1}_{\mathcal{C}_{j_1}}^T\mathbf{1}_{\mathcal{C}_{j_2}} - r\sum_{j_2=1}^{s}\mathbf{1}_{\mathcal{C}_{j_2}}^T\mathbf{1}_{\mathcal{C}_r}$$

$$= \sum_{j_1=1}^{r}\sum_{j_2=1}^{s}\mathbf{1}_{\mathcal{C}_{j_1}}^T\mathbf{1}_{\mathcal{C}_{j_2}} - r\mathbf{1}_{\mathcal{C}_r}^T\mathbf{1}_{\mathcal{C}_r}$$

$$= \sum_{j_1=1}^{r}\sum_{j_2=1}^{s}\left(\mathbf{1}_{\mathcal{C}_{j_1}}^T\mathbf{1}_{\mathcal{C}_{j_2}}\right) - r\,|\mathcal{C}|$$

$$= \sum_{j_1=1}^{r}\left(\mathbf{1}_{\mathcal{C}_{j_1}}^T\mathbf{1}_{\mathcal{C}_{j_1}}\right) - r\,|\mathcal{C}|$$

$$= \sum_{j_1=1}^{r}|\mathcal{C}| - r\,|\mathcal{C}|$$

$$= r\,|\mathcal{C}| - r\,|\mathcal{C}|$$

$$= 0$$

where in the third step we used that fact that $\mathbf{1}_{\mathcal{C}_r}^T\mathbf{1}_{\mathcal{C}_s} = 0$ as $r < s$, and $\mathbf{1}_{\mathcal{C}_{j_1}}^T\mathbf{1}_{\mathcal{C}_s} = 0$ as $j_1 < s$.

Finally, we can see that for $2 \leq r \leq k$

$$\chi_1^T\chi_r = \mathbf{1}^T\left(\sum_{j=1}^{r}\mathbf{1}_{\mathcal{C}_j} - r\mathbf{1}_{\mathcal{C}_r}\right)$$

$$= \sum_{j=1}^{r}\left(\mathbf{1}^T\mathbf{1}_{\mathcal{C}_j}\right) - r\mathbf{1}^T\mathbf{1}_{\mathcal{C}_r}$$

$$= r\,|\mathcal{C}| - r\,|\mathcal{C}|$$
$$= 0$$

and hence $\chi_1, \ldots, \chi_k$ are orthogonal eigenvectors of the matrix $\mathcal{W}$.

$\square$

The following Lemma shows the eigenvectors and eigenvalues of the power mean Laplacian in expectation under the considered Stochastic Block Model.

**Lemma 3.** *Let $E(\mathbb{G})$ be the expected multilayer graph with $T$ layers following the multilayer SBM with $k$ classes $\mathcal{C}_1, \ldots, \mathcal{C}_k$ of equal size and parameters $\left(p_{\mathrm{in}}^{(t)}, p_{\mathrm{out}}^{(t)}\right)_{t=1}^{T}$. Then the eigenvalues of the power mean Laplacian $\mathcal{L}_p$ are*

$$\lambda_1(\mathcal{L}_p) = \varepsilon, \qquad \lambda_i(\mathcal{L}_p) = m_p(\boldsymbol{\rho_\epsilon}), \qquad \lambda_j(\mathcal{L}_p) = 1 + \varepsilon \tag{13}$$

*with eigenvectors*

$$\chi_1 = \mathbf{1}, \qquad \chi_i = \sum_{j=1}^{i} \mathbf{1}_{\mathcal{C}_j} - i \mathbf{1}_{\mathcal{C}_i}$$

*where $(\boldsymbol{\rho_\epsilon})_t = 1 - (p_{\mathrm{in}}^{(t)} - p_{\mathrm{out}}^{(t)})/(p_{\mathrm{in}}^{(t)} + (k-1)p_{\mathrm{out}}^{(t)}) + \epsilon$, $t = 1, \ldots, T$, $i = 2, \ldots, k$, and $j = k+1, \ldots, |V|$,*

*Proof.* From Lemma 2 we know that $\chi_1, \ldots, \chi_k$ are eigenvectors of $\mathcal{W}^{(1)}, \ldots, \mathcal{W}^{(T)}$. In particular, we have seen that

$$\lambda_1^{(t)} = |\mathcal{C}|\,(p_{\mathrm{in}}^{(t)} + (k-1)p_{\mathrm{out}}^{(t)}), \quad \lambda_i^{(t)} = |\mathcal{C}|\,(p_{\mathrm{in}}^{(t)} - p_{\mathrm{out}}^{(t)})$$

for $i = 2, \ldots, k$. Further, as matrices $\mathcal{W}^{(1)}, \ldots, \mathcal{W}^{(T)}$ share all their eigenvectors, they are simultaneously diagonalizable, i.e. there exists a non-singular matrix $\Sigma$ such that $\Sigma^{-1}\mathcal{W}^{(t)}\Sigma = \Lambda^{(t)}$, where $\Lambda^{(t)}$ are diagonal matrices $\Lambda^{(t)} = \mathrm{diag}(\lambda_1^{(t)}, \ldots, \lambda_k^{(t)}, 0, \ldots, 0)$.

As we assume that all clusters are of the same size $|\mathcal{C}|$, the expected layer graphs are regular graphs with degrees $d^{(1)}, \ldots, d^{(T)}$. Hence, the normalized Laplacians of the expected layer graphs can be expressed as

$$\mathcal{L}_{\mathrm{sym}}^{(t)} = \Sigma(I - \frac{1}{d^{(t)}}\Lambda^{(t)})\Sigma^{-1}$$

Thus, we can observe that

$$\lambda_1^{(t)}(\mathcal{L}_{\mathrm{sym}}^{(t)}) = 0, \qquad \lambda_i^{(t)}(\mathcal{L}_{\mathrm{sym}}^{(t)}) = 1 - \rho^{(t)}, \qquad \lambda_j^{(t)}(\mathcal{L}_{\mathrm{sym}}^{(t)}) = 1,$$

for $i = 2, \ldots, k$, and $j = k+1, \ldots, |V|$, where

$$\rho^{(t)} = (p_{\mathrm{in}}^{(t)} - p_{\mathrm{out}}^{(t)})/(p_{\mathrm{in}}^{(t)} + (k-1)p_{\mathrm{out}}^{(t)})$$

for $t = 1, \ldots, T$. By obtaining the power mean Laplacian on diagonally shifted matrices,

$$\mathcal{L}_p = M_p(\mathcal{L}_{\mathrm{sym}}^{(1)} + \varepsilon I, \ldots, \mathcal{L}_{\mathrm{sym}}^{(1)} + \varepsilon I)$$

we have by Lemma 1

$$\lambda_1(\mathcal{L}_p) = m_p(\lambda_1^{(1)} + \varepsilon, \ldots, \lambda_1^{(T)} + \varepsilon) = \varepsilon$$
$$\lambda_i(\mathcal{L}_p) = m_p(1 - \rho^{(1)} + \varepsilon, \ldots, 1 - \rho^{(T)} + \varepsilon) = m_p(\boldsymbol{\rho_\epsilon}) \tag{14}$$
$$\lambda_j(\mathcal{L}_p) = m_p(\lambda_j^{(1)} + \varepsilon, \ldots, \lambda_j^{(T)} + \varepsilon) = 1 + \varepsilon$$

where $(\boldsymbol{\rho_\epsilon})_t = 1 - (p_{\mathrm{in}}^{(t)} - p_{\mathrm{out}}^{(t)})/(p_{\mathrm{in}}^{(t)} + (k-1)p_{\mathrm{out}}^{(t)}) + \epsilon$, and $t = 1, \ldots, T$, $i = 2, \ldots, k$, and $j = k+1, \ldots, |V|$, $\square$

The following Lemma describes the general form of the solution matrix

$$F = (f^{(1)}, \ldots, f^{(k)})$$

where the columns of $F$ are obtained from the following optimization problem

$$f^{(r)} = \arg\min_{f \in \mathbb{R}^n} \|f - CY^{(r)}\|^2 + \lambda f^T L_p f$$

Observe that this setting contains as a particular case the problem described in Eq. (1).

**Lemma 4.** *Let $E(\mathbb{G})$ be the expected multilayer graph with $T$ layers following the multilayer SBM with $k$ classes $\mathcal{C}_1, \ldots, \mathcal{C}_k$ of equal size and parameters $\left(p_{\mathrm{in}}^{(t)}, p_{\mathrm{out}}^{(t)}\right)_{t=1}^T$. Let $\rho_\epsilon$ be defined as in Lemma 3. Let $n_1, \ldots, n_k$ be the number of labeled nodes per class. Let $C \in \mathbb{R}^{n \times n}$ be a diagonal matrix with $C_{ii} = c_r$ for $v_i \in \mathcal{C}_r$. Let $l(v_i)$ be the label of node $v_i$, i.e. $l(v_i) = r$ if and only if $v_i \in \mathcal{C}_r$. Let the solution matrix $F = (f^{(1)}, \ldots, f^{(k)})$ where*

$$f^{(r)} = \arg\min_{f \in \mathbb{R}^n} \|f - CY^{(r)}\|^2 + \mu f^T \mathcal{L}_p f$$

*Then the solution matrix F is such that:*

- *If $r < l(v_i)$, then*

$$f_i^{(r)} = c_r \frac{n_r}{n} \alpha + c_r n_r \beta \left( (1 - l(v_i)) \frac{1}{\|\chi_{l(v_i)}\|^2} + \sum_{j=l(v_i)+1}^k \frac{1}{\|\chi_j\|^2} \right)$$

- *If $r > l(v_i)$, then*

$$f_i^{(r)} = c_r \frac{n_r}{n} \alpha + c_r n_r \beta \left( (1 - r) \frac{1}{\|\chi_r\|^2} + \sum_{j=r+1}^k \frac{1}{\|\chi_j\|^2} \right)$$

- *If $r = l(v_i)$, then*

$$f_i^{(r)} = c_r \frac{n_r}{n} \alpha + c_r n_r \beta \left( (1 - r)^2 \frac{1}{\|\chi_r\|^2} + \sum_{j=r+1}^k \frac{1}{\|\chi_j\|^2} \right)$$

*where $\alpha = \frac{1}{1+\mu\epsilon} - \frac{1}{1+\mu(1+\epsilon)}$, and $\beta = \frac{1}{1+\mu m_p(\rho_\epsilon)} - \frac{1}{1+\mu(1+\epsilon)}$.*

*Proof.* Let $U \in \mathbb{R}^{n \times n}$ be an orthonormal matrix such that $U = (u_1, u_2, \ldots, u_n)$, with $u_i = \chi_i / \|\chi_i\|$ for $i = 1, \ldots, k$, where $\chi_1, \ldots, \chi_k$ are eigenvectors of the power mean Laplacian as described in Lemma 3.

The power mean Laplacian $\mathcal{L}_p$ is a symmetric positive semidefinite matrix (see Lemma 3) and hence we can express $\mathcal{L}_p$ as $U\Lambda U^T$ where $\Lambda$ is a diagonal matrix with entries $\Lambda_{ii} = \lambda_i(\mathcal{L}_p)$, with $i = 1, \ldots, n$. Hence, we can see that

$$(I + \mu\mathcal{L}_p)^{-1} = (I + U\Lambda U^T)^{-1} = (U(I + \Lambda)U^T)^{-1} = U(I + \Lambda)^{-1}U^T = U\Omega U^T$$

where $\Omega$ is a diagonal matrix with entries $\Omega_{ii} = \frac{1}{1+\mu\lambda_i}$, with $i = 1, \ldots, n$.

From Lemma 3 we know that $\lambda_{k+1} = \cdots = \lambda_n = 1 + \epsilon =: \widehat{\omega}$, and hence it follows that $\Omega_{ii} = \frac{1}{1+\mu\widehat{\omega}}$ for $i = k + 1, \ldots, n$. Moreover, we can express $\Omega$ as the sum of two diagonal matrices, i.e.

$$\Omega = \omega I + \Theta$$

where $\omega = \frac{1}{1+\mu\widehat{\omega}}$ and $\Theta = \mathrm{diag}(\Omega_{11} - \omega, \ldots, \Omega_{kk} - \omega, 0, \ldots, 0)$. Observe that $\Theta_{11} = \Omega_{11} - \omega = \frac{1}{1+\mu\epsilon} - \frac{1}{1+\mu(1+\epsilon)} =: \alpha$ and $\Theta_{jj} = \Omega_{jj} - \omega = \frac{1}{1+\mu m_p(\rho_\epsilon)} - \frac{1}{1+\mu(1+\epsilon)} =: \beta$, for $j = 2, \ldots, k$.
Recall that we are interested in the equation

$$F = (I + \mu\mathcal{L}_p)^{-1}CY = U\Omega U^T CY \in \mathbb{R}^{n \times k},$$

where each column of $Y = [y^{(1)}, \ldots, y^{(k)}]$ is a class indicator of labeled nodes, i.e.

$$y_i^{(j)} = \begin{cases} 1 & \text{if } l(v_i) = j \\ 0 & \text{else} \end{cases} \tag{15}$$

Hence, each column of $Y$ can be expressed as

$$y^{(j)} = \sum_{v_i \in V | l(v_i) = j} e_i \tag{16}$$

where $e_i \in \mathbb{R}^n$ and $(e_i)_i = 1$ and zero else. With this in mind, we now study the matrix-vector product $U\Omega U^T e_i$. Recall that $U\Theta U^T$ is a $k$-rank matrix. Hence we have

$$
\begin{aligned}
U\Omega U^T e_i &= U(\omega I + \Theta) U^T e_i \\
&= \omega e_i + U\Theta U^T e_i \\
&= \omega e_i + \left( \sum_{j=1}^{k} \Theta_{jj} u_j u_j^T \right) e_i \\
&= \omega e_i + \left( \sum_{j=1}^{k} \frac{1}{\|\chi_j\|^2} \Theta_{jj} \chi_j \chi_j^T \right) e_i \\
&= \omega e_i + \frac{1}{n} \Theta_{11} \chi_1 + \left( \sum_{j=2}^{k} \frac{1}{\|\chi_j\|^2} \Theta_{jj} \chi_j \chi_j^T \right) e_i \\
&= \omega e_i + \frac{1}{n} \alpha \chi_1 + \beta \left( \sum_{j=2}^{k} \frac{1}{\|\chi_j\|^2} \chi_j \chi_j^T \right) e_i
\end{aligned}
$$

where in the last steps we used the fact that $\chi_1^T e_i = \mathbf{1}^T e_i = 1$, and define $\alpha = \Theta_{11}$ and $\beta = \Theta_{jj}$ due to the fact that $\Theta_{jj}$ are all equal for $j = 2, \ldots, k$.

The remaining terms $\chi_j \chi_j^T e_i$ depend on the cluster to which the corresponding node $v_i$ belongs to.

We first study the vector product $\chi_r^T e_i$. Observe that

$$\chi_r^T e_i = \left( \sum_{j=1}^{r} \mathbf{1}_{\mathcal{C}_j} - r \mathbf{1}_{\mathcal{C}_r} \right)^T e_i = \sum_{j=1}^{r} \left( \mathbf{1}_{\mathcal{C}_j}^T e_i \right) - r \mathbf{1}_{\mathcal{C}_r}^T e_i$$

Recall that $l(v_i)$ is the label of node $v_i$, i.e. $l(v_i) = r$ if and only if $v_i \in \mathcal{C}_r$. Then, we have

$$\sum_{j=1}^{r} \left( \mathbf{1}_{\mathcal{C}_j}^T e_i \right) - r \mathbf{1}_{\mathcal{C}_r}^T e_i = \begin{cases} 0 & \text{for } r < l(v_i) \\ 1 - l(v_i) & \text{for } r = l(v_i) \\ 1 & \text{for } r > l(v_i) \end{cases} \tag{17}$$

Therefore,

$$\left( \sum_{j=2}^{k} \frac{1}{\|\chi_j\|^2} \chi_j \chi_j^T \right) e_i = (1 - l(v_i)) \frac{\chi_{l(v_i)}}{\left\| \chi_{l(v_i)} \right\|^2} + \sum_{j=l(v_i)+1}^{k} \frac{\chi_j}{\|\chi_j\|^2}$$

All in all we have

$$U\Omega U^T e_i = \omega e_i + \frac{1}{n} \alpha \chi_1 + \beta \left( (1 - l(v_i)) \frac{\chi_{l(v_i)}}{\left\| \chi_{l(v_i)} \right\|^2} + \sum_{j=l(v_i)+1}^{k} \frac{\chi_j}{\|\chi_j\|^2} \right)$$

Moreover, the solution matrix $F$ can now be described column-wise as follows

$$f^{(r)} = (I + \mu\mathcal{L}_p)^{-1} C y^{(r)}$$

$$= c_r \left( \sum_{v_i \in V | l(v_i) = r} U\Omega U^T e_i \right)$$

$$= c_r \left( \sum_{v_i \in V | l(v_i) = r} \omega e_i \right) + \frac{1}{n} c_r n_r \alpha \boldsymbol{\chi}_1 + c_r n_r \beta \left( (1 - l(v_i)) \frac{\boldsymbol{\chi}_{l(v_i)}}{\left\| \boldsymbol{\chi}_{l(v_i)} \right\|^2} + \sum_{j=l(v_i)+1}^{k} \frac{\boldsymbol{\chi}_j}{\left\| \boldsymbol{\chi}_j \right\|^2} \right)$$

$$= \omega c_r y^{(r)} + c_r n_r \left( \frac{1}{n} \alpha \boldsymbol{\chi}_1 + \beta \left( (1 - r) \frac{\boldsymbol{\chi}_r}{\left\| \boldsymbol{\chi}_r \right\|^2} + \sum_{j=r+1}^{k} \frac{\boldsymbol{\chi}_j}{\left\| \boldsymbol{\chi}_j \right\|^2} \right) \right)$$

We now study the columns of matrix $F$. For this, observe that the $i^{th}$ entry of the column corresponding to the class $r$, is obtained by $f_i^{(r)} = \langle e_i, f^{(r)} \rangle$, and hence have

$$\langle e_i, f^{(r)} \rangle = \langle e_i, \omega c_r y^{(r)} + c_r n_r \left( \frac{1}{n} \alpha \boldsymbol{\chi}_1 + \beta \left( (1 - r) \frac{\boldsymbol{\chi}_r}{\left\| \boldsymbol{\chi}_r \right\|^2} + \sum_{j=r+1}^{k} \frac{\boldsymbol{\chi}_j}{\left\| \boldsymbol{\chi}_j \right\|^2} \right) \right) \rangle$$

$$= c_r \frac{n_r}{n} \alpha + c_r n_r \beta \langle e_i, \left( (1 - r) \frac{\boldsymbol{\chi}_r}{\left\| \boldsymbol{\chi}_r \right\|^2} c_r + \sum_{j=r+1}^{k} \frac{\boldsymbol{\chi}_j}{\left\| \boldsymbol{\chi}_j \right\|^2} \right) \rangle$$

where $\langle e_i, \omega c_r y^{(r)} \rangle = 0$ for unlabeled nodes. Having this, we now proceed to study three different cases of the remaining inner product. We do this by considering the following cases and making use of Eq. (17):

**First case:** $f_i^{(r)}$ **with** $r < l(v_i)$. We first analyze the following term

$$\langle e_i, \left( (1 - r) \frac{\boldsymbol{\chi}_r}{\left\| \boldsymbol{\chi}_r \right\|^2} + \sum_{j=r+1}^{k} \frac{\boldsymbol{\chi}_j}{\left\| \boldsymbol{\chi}_j \right\|^2} \right) \rangle = \langle e_i, (1 - r) \frac{\boldsymbol{\chi}_r}{\left\| \boldsymbol{\chi}_r \right\|^2} \rangle + \langle e_i, \sum_{j=r+1}^{k} \frac{\boldsymbol{\chi}_j}{\left\| \boldsymbol{\chi}_j \right\|^2} \rangle$$

$$\text{(by first case of Eq.17)} \qquad = \langle e_i, \sum_{j=r+1}^{k} \frac{\boldsymbol{\chi}_j}{\left\| \boldsymbol{\chi}_j \right\|^2} \rangle$$

$$\text{(by cases of Eq.17)} \qquad = (1 - l(v_i)) \frac{1}{\left\| \boldsymbol{\chi}_{l(v_i)} \right\|^2} + \sum_{j=l(v_i)+1}^{k} \frac{1}{\left\| \boldsymbol{\chi}_j \right\|^2}$$

Thus, we have

$$f_i^{(r)} = c_r \frac{n_r}{n} \alpha + c_r n_r \beta \left( (1 - l(v_i)) \frac{1}{\left\| \boldsymbol{\chi}_{l(v_i)} \right\|^2} + \sum_{j=l(v_i)+1}^{k} \frac{1}{\left\| \boldsymbol{\chi}_j \right\|^2} \right)$$

**Second case:** $f_i^{(r)}$ **with** $r > l(v_i)$. We first analyze the following term

$$\langle e_i, \left( (1 - r) \frac{\boldsymbol{\chi}_r}{\left\| \boldsymbol{\chi}_r \right\|^2} + \sum_{j=r+1}^{k} \frac{\boldsymbol{\chi}_j}{\left\| \boldsymbol{\chi}_j \right\|^2} \right) \rangle = \langle e_i, (1 - r) \frac{\boldsymbol{\chi}_r}{\left\| \boldsymbol{\chi}_r \right\|^2} \rangle + \langle e_i, \sum_{j=r+1}^{k} \frac{\boldsymbol{\chi}_j}{\left\| \boldsymbol{\chi}_j \right\|^2} \rangle$$

$$\text{(by third case of Eq.17)} \qquad = (1 - r) \frac{1}{\left\| \boldsymbol{\chi}_r \right\|^2} + \sum_{j=r+1}^{k} \frac{1}{\left\| \boldsymbol{\chi}_j \right\|^2}$$

Thus, we have

$$f_i^{(r)} = c_r \frac{n_r}{n}\alpha + c_r n_r \beta \left( (1-r)\frac{1}{\|\chi_r\|^2} + \sum_{j=r+1}^{k} \frac{1}{\|\chi_j\|^2} \right)$$

**Third case:** $f_i^{(r)}$ **with** $r = l(v_i)$. We first analyze the following term

$$\langle e_i, \left( (1-r)\frac{\chi_r}{\|\chi_r\|^2} + \sum_{j=r+1}^{k} \frac{\chi_j}{\|\chi_j\|^2} \right) \rangle = \langle e_i, (1-r)\frac{\chi_r}{\|\chi_r\|^2} \rangle + \langle e_i, \sum_{j=r+1}^{k} \frac{\chi_j}{\|\chi_j\|^2} \rangle$$

(by second case of Eq.17)
$$= (1-r)^2 \frac{1}{\|\chi_r\|^2} + \sum_{j=r+1}^{k} \frac{1}{\|\chi_j\|^2}$$

Thus, we have

$$f_i^{(r)} = c_r \frac{n_r}{n}\alpha + c_r n_r \beta \left( (1-r)^2 \frac{1}{\|\chi_r\|^2} + \sum_{j=r+1}^{k} \frac{1}{\|\chi_j\|^2} \right)$$

These three cases are the desired conditions.

$\square$

# B  Proof Of Theorem 1

**Theorem 5.** *Let $E(\mathbb{G})$ be the expected multilayer graph with $T$ layers following the multilayer SBM with $k$ classes $\mathcal{C}_1, \ldots, \mathcal{C}_k$ of equal size and parameters $\left( p_{\mathrm{in}}^{(t)}, p_{\mathrm{out}}^{(t)} \right)_{t=1}^{T}$. Let the same number of nodes per class be labeled. Then, a zero test classification error is achieved if and only if*

$$m_p(\boldsymbol{\rho_\epsilon}) < 1 + \epsilon,$$

*where $(\boldsymbol{\rho_\epsilon})_t = 1 - (p_{\mathrm{in}}^{(t)} - p_{\mathrm{out}}^{(t)})/(p_{\mathrm{in}}^{(t)} + (k-1)p_{\mathrm{out}}^{(t)}) + \epsilon$, and $t = 1, \ldots, T$.*

*Proof.* The proof of this theorem builds on top of Lemma 4, where the entries of the solution matrix $F = (f^{(1)}, \ldots, f^{(k)})$ are described, where

$$f^{(r)} = \underset{f \in \mathbb{R}^n}{\arg\min} \|f - CY^{(r)}\|^2 + \mu f^T \mathcal{L}_p f$$

Let $l(v_i)$ be the label of node $v_i$, i.e. $l(v_i) = r$ if and only if $v_i \in \mathcal{C}_r$. According to Lemma 4 the entries of matrix $F$ for unlabeled nodes are such that

- If $r < l(v_i)$, then

$$f_i^{(r)} = c_r \frac{n_r}{n}\alpha + c_r n_r \beta \left( (1-l(v_i))\frac{1}{\|\chi_{l(v_i)}\|^2} + \sum_{j=l(v_i)+1}^{k} \frac{1}{\|\chi_j\|^2} \right)$$

- If $r > l(v_i)$, then

$$f_i^{(r)} = c_r \frac{n_r}{n}\alpha + c_r n_r \beta \left( (1-r)\frac{1}{\|\chi_r\|^2} + \sum_{j=r+1}^{k} \frac{1}{\|\chi_j\|^2} \right)$$

- If $r = l(v_i)$, then

$$f_i^{(r)} = c_r \frac{n_r}{n}\alpha + c_r n_r \beta \left( (1-r)^2 \frac{1}{\|\chi_r\|^2} + \sum_{j=r+1}^{k} \frac{1}{\|\chi_j\|^2} \right)$$

where $\alpha = \frac{1}{1+\mu\epsilon} - \frac{1}{1+\mu(1+\epsilon)}$, and $\beta = \frac{1}{1+\mu m_p(\boldsymbol{\rho_\epsilon})} - \frac{1}{1+\mu(1+\epsilon)}$.

Observe that the case here considered corresponds to the case where the amount of labeled data per class is the same, i.e. $n_1 = \cdots = n_k$, and where the matrix $C$ is the identity, i.e. $c_1 = \cdots c_r = 1$.

Moreover, the estimated label assignment for unlabeled nodes goes by the following rule

$$\hat{l}(v_i) = \arg\max\{f_i^{(1)}, \ldots, f_i^{(k)}\}$$

Hence, we need to find conditions so that the following inequality holds

$$f_i^{(j)} < f_i^{(l(v_i))} \qquad \forall\, j \neq l(v_i)$$

Hence, we consider the following two cases:

**Case 1:** $f_i^{(r)} < f_i^{(l(v_i))}$ **for** $r > l(v_i)$.

Let $r^* = l(v_i)$, and $r = r^* + \Delta$. Then, we have

$$f_i^{(r)} < f_i^{(l(v_i))} \Leftrightarrow$$
$$f_i^{(r)} < f_i^{(r^*)} \Leftrightarrow$$
$$\beta\left((1-r)\frac{1}{\|\boldsymbol{\chi}_r\|^2} + \sum_{j=r+1}^{k}\frac{1}{\|\boldsymbol{\chi}_j\|^2}\right) < \beta\left((1-r^*)^2\frac{1}{\|\boldsymbol{\chi}_{r^*}\|^2} + \sum_{j=r^*+1}^{k}\frac{1}{\|\boldsymbol{\chi}_j\|^2}\right) \Leftrightarrow$$
$$0 < \beta\left((1-r^*)^2\frac{1}{\|\boldsymbol{\chi}_{r^*}\|^2} - (1-r)\frac{1}{\|\boldsymbol{\chi}_r\|^2} + \sum_{j=r^*+1}^{k}\frac{1}{\|\boldsymbol{\chi}_j\|^2} - \sum_{j=r+1}^{k}\frac{1}{\|\boldsymbol{\chi}_j\|^2}\right) \Leftrightarrow$$
$$0 < \beta\left((1-r^*)^2\frac{1}{\|\boldsymbol{\chi}_{r^*}\|^2} + (r-1)\frac{1}{\|\boldsymbol{\chi}_r\|^2} + \sum_{j=r^*+1}^{k}\frac{1}{\|\boldsymbol{\chi}_j\|^2} - \sum_{j=r^*+\Delta+1}^{k}\frac{1}{\|\boldsymbol{\chi}_j\|^2}\right) \Leftrightarrow$$
$$0 < \beta\left((1-r^*)^2\frac{1}{\|\boldsymbol{\chi}_{r^*}\|^2} + (r-1)\frac{1}{\|\boldsymbol{\chi}_r\|^2} + \sum_{j=r^*+1}^{r^*+\Delta}\frac{1}{\|\boldsymbol{\chi}_j\|^2}\right) \Leftrightarrow$$
$$0 < \beta$$

**Case 2:** $f_i^{(r)} < f_i^{(l(v_i))}$ **for** $r < l(v_i)$.

Let $r^* = l(v_i)$, and $r^* = r + \Delta$. Then, we have

$$f_i^{(r)} < f_i^{(l(v_i))} \Leftrightarrow$$
$$f_i^{(r)} < f_i^{(r^*)} \Leftrightarrow$$
$$\beta\left((1-r^*)\frac{1}{\|\boldsymbol{\chi}_{r^*}\|^2} + \sum_{j=r^*+1}^{k}\frac{1}{\|\boldsymbol{\chi}_j\|^2}\right) < \beta\left((1-r^*)^2\frac{1}{\|\boldsymbol{\chi}_{r^*}\|^2} + \sum_{j=r^*+1}^{k}\frac{1}{\|\boldsymbol{\chi}_j\|^2}\right) \Leftrightarrow$$
$$0 < \beta\left((1-r^*)^2\frac{1}{\|\boldsymbol{\chi}_{r^*}\|^2} - (1-r^*)\frac{1}{\|\boldsymbol{\chi}_{r^*}\|^2}\right)$$
$$0 < \beta\left((1-r^*)^2\frac{1}{\|\boldsymbol{\chi}_{r^*}\|^2} + (r^*-1)\frac{1}{\|\boldsymbol{\chi}_{r^*}\|^2}\right) \Leftrightarrow$$
$$0 < \beta$$

All in all, from the two considered cases we can see that

$$f_i^{(j)} < f_i^{(l(v_i))} \qquad \forall\, j \neq l(v_i) \iff 0 < \beta$$

In fact,

$$0 < \beta \Leftrightarrow$$

$$0 < \frac{1}{1 + \mu m_p(\boldsymbol{\rho_\epsilon})} - \frac{1}{1 + \mu(1 + \epsilon)} \Leftrightarrow$$

$$\frac{1}{1 + \mu(1 + \epsilon)} < \frac{1}{1 + \mu m_p(\boldsymbol{\rho_\epsilon})} \Leftrightarrow$$

$$1 + \mu m_p(\boldsymbol{\rho_\epsilon}) < 1 + \mu(1 + \epsilon) \Leftrightarrow$$

$$m_p(\boldsymbol{\rho_\epsilon}) < 1 + \epsilon$$

which is the desired condition. □

## C  Proof of Corollary 1

**Corollary 3.** *Let $E(\mathbb{G})$ be an expected multilayer graph as in Theorem 1. Then,*

- *For $p \to \infty$, the classification error is zero if and only if $p_{\text{out}}^{(t)} < p_{\text{in}}^{(t)}$ for all $t = 1, \ldots, T$.*
- *For $p \to -\infty$, the classification error is zero if and only there exists a $t \in \{1, \ldots, T\}$ s.t. $p_{\text{out}}^{(t)} < p_{\text{in}}^{(t)}$.*

*Proof.* Observe that the limit cases of the scalar power means are

$$\lim_{p \to -\infty} m_p(x_1, \ldots, x_T) = \min\{x_1, \ldots, x_T\}$$

$$\lim_{p \to +\infty} m_p(x_1, \ldots, x_T) = \max\{x_1, \ldots, x_T\}$$

Applying this to condition

$$m_p(\boldsymbol{\rho_\epsilon}) < 1 + \epsilon,$$

where $(\boldsymbol{\rho_\epsilon})_t = 1 - (p_{\text{in}}^{(t)} - p_{\text{out}}^{(t)})/(p_{\text{in}}^{(t)} + (k - 1)p_{\text{out}}^{(t)}) + \epsilon$, and $t = 1, \ldots, T$ yields the desired result. □

## D  Proof of Corollary 2

**Corollary 4.** *Let $E(\mathbb{G})$ be an expected multilayer graph as in Theorem 1. Let $p \leq q$. If $\mathcal{L}_q$ has a zero-classification error, then $\mathcal{L}_p$ has a zero-classification error.*

*Proof.* By Theorem 4 we have that if $p \leq q$ then $m_p(x_1, \ldots, x_T) \leq m_p(x_1, \ldots, x_T)$. Therefore, applying this to our case we can see that

$$m_p(\boldsymbol{\rho_\epsilon}) \leq m_q(\boldsymbol{\rho_\epsilon}) < 1 + \epsilon$$

A zero test classification error with parameter $q$ is achieved if and only if $m_q(\boldsymbol{\rho_\epsilon}) < 1 + \epsilon$, hence we can see that zero test classification error with parameter $p$ is achieved if it is achieved with parameter $q$ and $p \leq q$.

□

## E  General version of Theorem 2

**Theorem 6.** *Let $E(\mathbb{G})$ be the expected multilayer graph with $T$ layers following the multilayer SBM with two classes $\mathcal{C}_1, \mathcal{C}_2$ of equal size and parameters $\left( p_{\text{in}}^{(t)}, p_{\text{out}}^{(t)} \right)_{t=1}^T$. Let $n_1, n_2$ nodes from classes $\mathcal{C}_1, \mathcal{C}_2$ be labeled, respectively. Let $\mu = 1$. Then, a zero test classification error is achieved if and only if*

$$m_p(\boldsymbol{\rho_\epsilon}) < \min \left\{ \frac{(n_1 + n_2)((1 + \epsilon)^2 + 1) - 2n_2}{2n_2 + (n_1 + n_2)\epsilon}, \frac{(n_1 + n_2)((1 + \epsilon)^2 + 1) - 2n_1}{2n_1 + (n_1 + n_2)\epsilon} \right\}$$

*where $(\boldsymbol{\rho_\epsilon})_l = 1 - (p_{\text{in}}^{(l)} - p_{\text{out}}^{(l)})/(p_{\text{in}}^{(l)} + (k - 1)p_{\text{out}}^{(l)}) + \epsilon$, and $l = 1, \ldots, T$.*

*Proof.* The proof of this theorem builds on top of Lemma 4, where the entries of the solution matrix $F = (f^{(1)}, \ldots, f^{(k)})$ are described, where

$$f^{(r)} = \underset{f \in \mathbb{R}^n}{\arg\min} \| f - CY^{(r)} \|^2 + \mu f^T \mathcal{L}_p f$$

Let $l(v_i)$ be the label of node $v_i$, i.e. $l(v_i) = r$ if and only if $v_i \in \mathcal{C}_r$. According to Lemma 4 the entries of matrix $F$ for unlabeled nodes are such that

- If $r < l(v_i)$, then

$$f_i^{(r)} = c_r \frac{n_r}{n} \alpha + c_r n_r \beta \left( (1 - l(v_i)) \frac{1}{\|\boldsymbol{\chi}_{l(v_i)}\|^2} + \sum_{j=l(v_i)+1}^{k} \frac{1}{\|\boldsymbol{\chi}_j\|^2} \right)$$

- If $r > l(v_i)$, then

$$f_i^{(r)} = c_r \frac{n_r}{n} \alpha + c_r n_r \beta \left( (1 - r) \frac{1}{\|\boldsymbol{\chi}_r\|^2} + \sum_{j=r+1}^{k} \frac{1}{\|\boldsymbol{\chi}_j\|^2} \right)$$

- If $r = l(v_i)$, then

$$f_i^{(r)} = c_r \frac{n_r}{n} \alpha + c_r n_r \beta \left( (1 - r)^2 \frac{1}{\|\boldsymbol{\chi}_r\|^2} + \sum_{j=r+1}^{k} \frac{1}{\|\boldsymbol{\chi}_j\|^2} \right)$$

where $\alpha = \frac{1}{1+\mu\epsilon} - \frac{1}{1+\mu(1+\epsilon)}$, and $\beta = \frac{1}{1+\mu m_p(\boldsymbol{\rho}_\epsilon)} - \frac{1}{1+\mu(1+\epsilon)}$.

Observe that the case here considered corresponds to the case with two classes, i.e. $k = 2$ with equal size classes $\mathcal{C}_1$ and $\mathcal{C}_2$ where the amount of labeled data per class is $n_1$ and $n_2$, respectively, with the matrix $C$ as the identity, i.e. $c_1 = c_2 = 1$, and regularization parameter $\mu = 1$.

Moreover, the estimated label assignment for unlabeled nodes goes by the following rule

$$\hat{l}(v_i) = \arg\max\{f_i^{(1)}, f_i^{(2)}\}$$

Hence, we need to find conditions so that the following inequality holds

$$f_i^{(j)} < f_i^{(l(v_i))} \qquad \forall\, j \neq l(v_i)$$

Let $l(v_i) = 1 \Leftrightarrow v_i \in \mathcal{C}_1$, and $l(v_i) = 2 \Leftrightarrow v_i \in \mathcal{C}_2$. A quick computation following Lemma 4 yields

- $f_i^{(1)} = \frac{n_1}{n} \alpha + n_1 \beta (\frac{1}{\|\boldsymbol{\chi}_2\|^2})$ for $v_i \in \mathcal{C}_1$, i.e. $l(v_i) = 1$

- $f_i^{(1)} = \frac{n_1}{n} \alpha - n_1 \beta (\frac{1}{\|\boldsymbol{\chi}_2\|^2})$ for $v_i \in \mathcal{C}_2$, i.e. $l(v_i) = 2$

- $f_i^{(2)} = \frac{n_2}{n} \alpha - n_2 \beta (\frac{1}{\|\boldsymbol{\chi}_2\|^2})$ for $v_i \in \mathcal{C}_1$, i.e. $l(v_i) = 1$

- $f_i^{(2)} = \frac{n_2}{n} \alpha + n_2 \beta (\frac{1}{\|\boldsymbol{\chi}_2\|^2})$ for $v_i \in \mathcal{C}_2$, i.e. $l(v_i) = 2$

Observing that $\|\boldsymbol{\chi}_2\|^2 = n$ these conditions can be rephrase as follows

$$f^{(1)} = \frac{n_1}{n} \left( (\alpha + \beta) \mathbf{1}_\mathcal{C} + (\alpha - \beta) \mathbf{1}_{\overline{\mathcal{C}}} \right)$$

$$f^{(2)} = \frac{n_2}{n} \left( (\alpha - \beta) \mathbf{1}_\mathcal{C} + (\alpha + \beta) \mathbf{1}_{\overline{\mathcal{C}}} \right)$$

Hence, the conditions for correct label assignment of unlabeled nodes are

$$n_1 (\alpha + \beta) > n_2 (\alpha - \beta) \text{ and } n_2 (\alpha + \beta) > n_1 (\alpha - \beta)$$

Let $\Omega_{11} = \frac{1}{1+\epsilon}$, $\Omega_{22} = \frac{1}{1+m_p(\boldsymbol{\rho}_\epsilon)}$, and $\omega = \frac{1}{1+(1+\epsilon)}$. Then, $\alpha = \Omega_{11} - \omega$, and $\beta = \Omega_{22} - \omega$.

By studying the first condition we observe

$$n_1 (\alpha + \beta) > n_2 (\alpha - \beta) \Leftrightarrow$$

$$n_1 (\Omega_{11} - \omega + \Omega_{22} - \omega) > n_2 (\Omega_{11} - \omega - (\Omega_{22} - \omega)) \Leftrightarrow$$

$$n_1 (\Omega_{11} + \Omega_{22} - 2\omega) > n_2(\Omega_{11} - \Omega_{22}) \Leftrightarrow$$

$$(n_1 - n_2)\Omega_{11} + (n_1 + n_2)\Omega_{22} > 2n_1\omega \Leftrightarrow$$

$$\Omega_{22} > \frac{1}{n_1 + n_2} (2n_1\omega - (n_1 - n_2)\Omega_{11}) \Leftrightarrow$$

$$\frac{1}{1 + m_p(\boldsymbol{\rho_\epsilon})} > \frac{1}{n_1 + n_2} \left(2n_1 \frac{1}{1 + (1 + \epsilon)} - (n_1 - n_2)\Omega_{11}\right) \Leftrightarrow$$

$$\frac{1}{1 + m_p(\boldsymbol{\rho_\epsilon})} > \frac{1}{n_1 + n_2} \left(2n_1 \frac{1}{2 + \epsilon} - (n_1 - n_2)\frac{1}{1 + \epsilon}\right) \Leftrightarrow$$

$$\frac{1}{1 + m_p(\boldsymbol{\rho_\epsilon})} > \frac{1}{n_1 + n_2} \left(\frac{2n_2 + (n_1 + n_2)\epsilon}{(2 + \epsilon)(1 + \epsilon)}\right) \Leftrightarrow$$

$$1 + m_p(\boldsymbol{\rho_\epsilon}) < (n_1 + n_2) \left(\frac{(2 + \epsilon)(1 + \epsilon)}{2n_2 + (n_1 + n_2)\epsilon}\right) \Leftrightarrow$$

$$m_p(\boldsymbol{\rho_\epsilon}) < (n_1 + n_2) \left(\frac{(2 + \epsilon)(1 + \epsilon)}{2n_2 + (n_1 + n_2)\epsilon}\right) - 1 \Leftrightarrow$$

$$m_p(\boldsymbol{\rho_\epsilon}) < (n_1 + n_2) \left(\frac{(2 + \epsilon)(1 + \epsilon) - (2n_2 + (n_1 + n_2)\epsilon)}{2n_2 + (n_1 + n_2)\epsilon}\right) \Leftrightarrow$$

$$m_p(\boldsymbol{\rho_\epsilon}) < \frac{(n_1 + n_2)((2 + \epsilon)(1 + \epsilon) - \epsilon) - 2n_2}{2n_2 + (n_1 + n_2)\epsilon} \Leftrightarrow$$

$$m_p(\boldsymbol{\rho_\epsilon}) < \frac{(n_1 + n_2)((1 + \epsilon)^2 + 1) - 2n_2}{2n_2 + (n_1 + n_2)\epsilon} \Leftrightarrow$$

The corresponding condition for $\mathcal{C}_2$ can be obtained in a similar way, yielding

$$m_p(\boldsymbol{\rho_\epsilon}) < \frac{(n_1 + n_2)((1 + \epsilon)^2 + 1) - 2n_1}{2n_1 + (n_1 + n_2)\epsilon}$$

Hence, both conditions hold if and only if

$$m_p(\boldsymbol{\rho_\epsilon}) = m_p(\boldsymbol{\rho_\epsilon}) < \min \left\{ \frac{(n_1 + n_2)((1 + \epsilon)^2 + 1) - 2n_2}{2n_2 + (n_1 + n_2)\epsilon}, \frac{(n_1 + n_2)((1 + \epsilon)^2 + 1) - 2n_1}{2n_1 + (n_1 + n_2)\epsilon} \right\}$$

$$\square$$

## F Proof of Theorem 2

**Theorem 7.** *Let $E(\mathbb{G})$ be the expected multilayer graph with $T$ layers following the multilayer SBM with two classes $\mathcal{C}_1, \mathcal{C}_2$ of equal size and parameters $\left(p_{\text{in}}^{(t)}, p_{\text{out}}^{(t)}\right)_{t=1}^{T}$. Let $n_1, n_2$ nodes from clusters $\mathcal{C}_1, \mathcal{C}_2$ be labeled, respectively. Let $\mu = 1$. Then, a zero test classification error is achieved if*

$$m_p(\boldsymbol{\rho_\epsilon}) < \min \left\{ \frac{n_1}{n_2}, \frac{n_2}{n_1} \right\}$$

*where $(\boldsymbol{\rho_\epsilon})_l = 1 - (p_{\text{in}}^{(l)} - p_{\text{out}}^{(l)})/(p_{\text{in}}^{(l)} + (k - 1)p_{\text{out}}^{(l)}) + \epsilon$, and $l = 1, \ldots, T$.*

*Proof.* We first analyze the first condition of the right hand side of Theorem 6. Let $g(\epsilon) = \frac{(n_1+n_2)((1+\epsilon)^2+1)-2n_2}{2n_2+(n_1+n_2)\epsilon}$. Then,

$$g(0) = \frac{2(n_1 + n_2) - 2n_2}{2n_2} = \frac{n_1}{n_2}$$

Moreover, it is clear that $g$ is monotone, as it is quadratic on $\epsilon$ on the numerator and linear on the denominator, and hence $g(0) < g(\epsilon)$.

A similar procedure with the second condition of the right hand side of Theorem 6 leads to the condition $\frac{n_2}{n_1}$, leading to the desired result. $\qquad\square$

## G  Proof of Theorem 3

**Theorem 8.** *Let $E(\mathbb{G})$ be the expected multilayer graph with $T$ layers following the multilayer SBM with $k$ classes $\mathcal{C}_1, \ldots, \mathcal{C}_k$ of equal size and parameters $\left(p_{\text{in}}^{(t)}, p_{\text{out}}^{(t)}\right)_{t=1}^{T}$. Let $n_1, \ldots, n_k$ be the number of nodes per class be labeled. Let $C \in \mathbb{R}^n$ be a cost vector where with $C_i = n/n_r$ for $v_i \in \mathcal{C}_r$. Then, a zero test classification error is achieved if and only if*

$$m_p(\boldsymbol{\rho_\epsilon}) < 1 + \epsilon \,,$$

*where $(\boldsymbol{\rho_\epsilon})_l = 1 - (p_{\text{in}}^{(l)} - p_{\text{out}}^{(l)})/(p_{\text{in}}^{(l)} + (k-1)p_{\text{out}}^{(l)}) + \epsilon$, and $l = 1, \ldots, T$.*

*Proof.* The proof is similar to the one of Theorem 1 (see Section B). The only change is in the terms $c_r \frac{n_r}{n}$. Since we have by definition that $c_r = \frac{n}{n_r}$ we have that $c_r \frac{n_r}{n} = 1$, leading to the conditions obtained by Theorem 1. $\qquad\square$

## H  A scalable matrix-free method for the linear system $(I + \lambda L_p)f = Y$

Computing the generalized matrix mean of $T$ positive definite matrices $A_1, \ldots, A_T$ requires to compute $T + 1$ matrix functions: $A_1^p, \ldots, A_T^p$ and $(\sum_i A_i^p)^{1/p}$. Typically, the matrices $A_i^p$ are full even though each $A_i$ is a sparse matrix and so, computing $L_p$ explicitly is unfeasible if the $A_i$'s have large dimensions. Given a vector $\mathbf{y}$ and a negative integer $p$, here we propose a matrix-free method for solving the linear system $(I + \lambda L_p)^{-1}\mathbf{y}$. The method exploits the sparsity of the Laplacians of each layer and is matrix-free in the sense that it requires only to compute the matrix-vector product $A_i \times vector$, without requiring to store the matrices $A_i$ themselves nor to compute any matrix function $A_i^p$ explicitly. Thus, when the layers are sparse, the method scales to large datasets. Below we give further details about the method presented in the short version of the paper. We present the method for a general set of positive definite matrices $A_1, \ldots, A_T$, and for a general vector $\mathbf{y}$, for the sake of generality.

Let $S_p = A_1^p + \cdots + A_T^p$, $\varphi : \mathbb{C} \to \mathbb{C}$ be the complex function $\varphi(z) = z^{1/p}$ and let $L_p$ be the matrix function $L_p = T^{-1/p}\varphi(S_p)$. The proposed method essentially transforms the original problem into a series of subproblems which thus allow us to solve the linear system $(I + \lambda L_p)^{-1}\mathbf{y}$ by solving several different linear systems with $A_i$ as coefficient matrices. The method consists of three main nested inner–steps which we present below.

**1.** First, we solve the linear system $(I + \lambda L_p)^{-1}\mathbf{y}$ by a Krylov method (GMRES in our case [27]). At each iteration, this method projects the problem into the Krylov subspace spanned by $\{\mathbf{y}, \lambda L_p\mathbf{y}, (\lambda L_p)^2\mathbf{y}, \ldots, (\lambda L_p)^h\mathbf{y}\}$. If $\kappa = \lambda_{\max}(L_p)/\lambda_{\min}(L_p)$, then the method converges as

$$O\left(\left(\frac{\kappa^2 - 1}{\kappa^2}\right)^{h/2}\right) \,.$$

Thus, if $L_p$ is well conditioned, a relatively small $h$ is required. In order to build the appropriate Krylov subspace, at each iteration we need to efficiently perform one matrix–vector product $L_p\mathbf{y}$.

**2.** Second, in order to compute $L_p\mathbf{y} = T^{-1/p}\varphi(S_p)\mathbf{y}$ we use the Cauchy integral form of the function $\varphi$, transformed via a conformal map, to approximate $\varphi(S_p)$ via the trapezoidal rule, as proposed in [12]. Let $m, M > 0$ be such that the interval $[m, M]$ contains the whole spectrum of $S_p$ and let $t_1, \ldots, t_N$ be $N$ equally spaced contour points to be used in the trapezoidal rule. As $\varphi$ has a singularity at $z = 0$ but just a brunch cut on $(-\infty, 0)$, we can approximate $\varphi(S_p)\mathbf{y}$ via [12]

$$\varphi_N(S_p)\mathbf{y} = \frac{-8K(mM)^{1/4}}{\pi N k} S_p \; \text{Im} \left\{ \sum_{i=1}^{N} \frac{\varphi(z_i^2)c_i d_i}{z_i(k^{-1} - s_i)^2}(z_i^2 I - S_p)^{-1}\mathbf{y} \right\}$$

where Im denotes the imaginary part, $k = \left((M/m)^{1/4} - 1\right)/\left((M/m)^{1/4} + 1\right)$, $K$ is the value of the complete elliptic integral of the first kind, evaluated at $ke^2$, $s_i = \mathrm{sn}(t_i)$ is the Jacobi elliptic sine function evaluated on the $i$-th contour point $t_i$, and

$$z_i = (mM)^{1/4}\left(\frac{k^{-1} + s_i}{k^{-1} - s_i}\right), \quad c_i = \sqrt{1 - s_i^2}, \quad d_i = \sqrt{1 - k^2 s_i^2},$$

for $i = 1, \ldots, N$. This approximation converges geometrically as the number of points increases. Precisely, it holds

$$\|\varphi(S_p)\mathbf{y} - \varphi_N(S_p)\mathbf{y}\| = O(e^{-2\pi^2 N/(\ln(M/m)+6)}).$$

Thus, the computation of $\varphi(S_p)\mathbf{y}$ is reduced to $N$ linear systems $(z_i^2 I - S_p)^{-1}\mathbf{y}$. Note that these systems are independent and thus they can be solved in parallel.

**3.** Finally, in order to solve the linear system $(zI - S_p)^{-1}\mathbf{y}$ we employ again a Krylov method. In order to build the Krylov space for $(zI - S_p)$ and $\mathbf{y}$ we need to efficiently perform one multiplication $S_p$ times a vector per iteration. As $S_p = \sum_{i=1}^{T} A_i^p = \sum_{i=1}^{T}(A_i^{-1})^{|p|}$, this problem reduces to solving $q$ linear systems with $A_i$ as coefficient matrix, for $i = 1, \ldots, T$. As the matrices $A_i$ are assumed sparse and positive definite, we can very efficiently solve each of these systems via the Preconditioned Conjugate Gradient method with an incomplete Cholesky preconditioner.

The pseudocode for the proposed algorithm is presented in Algorithms 1–3.

---

**Input:** $A_1, \ldots, A_T, p, \mathbf{y}, \lambda$
1 Compute preconditioners $P_1, \ldots, P_T$ for $A_1, \ldots, A_T$
2 Compute estimates for $m$ and $M$ such that eigenvalues$(S_p) \subseteq [m, M]$
3 Choose number of contour points $N$
4 Compute contour coefficients $z_i, s_i, K, k$
5 Solve $(I + \lambda L_p)^{-1}\mathbf{y}$ with GMRES, using Alg.2 as subroutine
   **Output:** $\mathbf{u} = (I + \lambda L_p)^{-1}\mathbf{y}$

**Algorithm 1:** Solve $(I + \lambda L_p)^{-1}\mathbf{y}$

---

**Input:** $A_1, \ldots, A_T, p, \mathbf{y}, N, m, M$, contour coefficients $z_i, s_i, c_i, d_i, k, K$
1 $\mathbf{u} \leftarrow S_p\mathbf{y}$, using Alg.3
2 **for** $i = 1, \ldots, N$ **do**
3    $\mathbf{u} \leftarrow \mathrm{solve}(z_i I - S_p, \mathbf{y})$ with GMRES, using Alg.3 as subroutine
4    $\mathbf{u} \leftarrow \frac{(z_i^2)^{1/p} c_i d_i}{z_i(k^{-1} - s_i)^2}\mathbf{u}$
5    $\mathbf{u}_{k+1} = \|\mathbf{v}_{k+1}\|_q^{1-q}|\mathbf{v}_{k+1}|^{q-2}\mathbf{v}_{k+1}$
6 **end**
7 $\mathbf{u} \leftarrow \frac{1}{T^{1/p}}\frac{-8K(mM)^{1/4}}{\pi N k}\mathrm{Im}(\mathbf{u})$
   **Output:** $\mathbf{u} = L_p\mathbf{y}$

**Algorithm 2:** Multiply $L_p$ times a vector

**Input:** $A_1, \ldots, A_T, P_1, \ldots, P_T, \mathbf{y}$
1 **for** $k = 1, \ldots, T$ **do**
2    $\mathbf{u} \leftarrow \mathbf{u} + \mathrm{solve}(A_i^{|p|}, \mathbf{y})$ using CG preconditioned with $P_i$
3 **end**
   **Output:** $\mathbf{u} = S_p\mathbf{y}$

**Algorithm 3:** Multiply $S_p$ times a vector

---

### H.1 Implementation details and computational complexity

Few implementation details are in order:

The preconditioners $P_i$ can be computed using an incomplete Cholesky factorization. In our test we observe that a `1e-4` threshold is enough to ensure convergence of Alg.3 to `1e-8` precision in just 2 or 3 iterations. As in our case the $A_i$ are Laplacians, another excellent preconditioner can be obtained using a Combinatorial Multi Grid method (CMG). In our experiments, the CMG preconditioner performed similarly (but slightly worse) than the incomplete Cholesky.

A precise estimate of $M$ in Alg.1 step 2 can be obtained using a Krylov eigensolver with Alg.3 as subroutine. As for $m$, since each $A_i^p$ is positive definite and $p$ is a negative integer, a good estimate

Figure 6: Experiments with three Layers and three classes as in Section 4.2 and Figure.4.

can be obtained by exploiting the Weyl's inequality (see e.g. [32])

$$m = \lambda_{\max}(A_1)^p + \cdots + \lambda_{\max}(A_T)^p \leq \lambda_{\min}(S_p).$$

The number of contour points $N$ can be chosen using the geometric convergence of $\varphi_N$. In our experiments, we chose a precision $\tau =$`1e-8` and we set

$$N = |(\ln(M/m) + 6)\ln(\tau)/2\pi^2|.$$

The contour points have been calculated using the code from [7].

Concerning the computational cost of the method, the following analysis shows that it is proportional to the number of edges in each layer, i.e. Alg.1 scales to large sparse datasets. Let $c(A_i)$ be the cost of multiplying $c(A_i)$ times a vector (which is proportional to the number of nonzeros in $A_i$, i.e. the number of edges in the layer $i$ when $A_i$ is the normalized Laplacian of the $i$-th layer). Let $K_1, K_2, K_3$ be the number of iterations of GMRES,GMRES and PCG in lines 5, 3 and 2 of Algorithms 1, 2 and 3, respectively. Each instance of solve($A_i^{|p|}, \mathbf{y}$) in Alg.3 requires $K_3 p\, c(A_i)$ operations per step. So The cost of Alg.3 is roughly $pK_3 \sum_{i=1}^T c(A_i)$. This implies that the cost of Alg.2 is $NK_2K_3p \sum_{i=1}^T c(A_i)$. Therefore, the cost of solving the linear system $(I + \lambda L_p)^{-1}\mathbf{y}$ with Alg.1 is

$$K_1 N K_2 K_3 p\big(c(A_1) + \cdots + c(A_T)\big),$$

showing that the method scales as the number of nonzeros in each layer, as claimed. It is important to notice that the Algorithm allows for a high level of parallelism. In fact, the computation of the preconditioners $P_i$ at step 1 of Alg.1, the **for** at step 2 of Alg.2 and the **for** at step 1 of Alg.3 can all be run in parallel.

# I   Analysis on Three Layers with Three Classes

In this section we give a more detailed exposition of experiments presented in Section 4.2. We consider the cases where $p_{\text{in}} - p_{\text{out}} \in \{0.03, 0.04, \dots, 0.1\}$ which are depicted in Fig.6. In the $x$-axis we have the amount of labeled nodes and in the $y$- we have the classification error. We can see that in general there is a trend between the performance of our proposed method (colorful curves) and state of the art methods (black curves). We can see that the larger the gap $p_{\text{in}} - p_{\text{out}}$ the larger the difference is between our proposed method and state of the art methods. Moreover, one can see that the smaller the value of $p$ the better the performance of our proposed method. Moreover, there is a set of state of the art methods that do not improve their performance with larger amounts of labeled nodes. Yet, one can observe that there are three methods from the state of the art that perform close to our methods: TLMV, ZooBP and AGML, which performs similarly to our method $L_1$ (i.e. the arithmetic mean of Laplacians).

Figure 7: Mean test classification error under MSBM for different values of $\lambda$. Details in Sec. J.

## J  Analysis on Effect of Regularization Parameter

In this section we present a numerical evaluation on the effect of the regularization parameter $\lambda$ under the multilayer stochastic block model and on real world datasets. The corresponding results are depicted in Fig. 7 and Fig. .

**Experiments under Multilayer Stochastic Block Model**. We analyze the effect of the regularization parameter $\lambda$ under the Multilayer Stochastic Block Model. The experimental setting is as follows: We fix the parameters of the first layer $G^{(1)}$ and second layer $G^{(2)}$ to $p_{\text{in}}^{(1)} = 0.09, p_{\text{out}}^{(1)} = 0.01, p_{\text{in}}^{(2)} = 0.05, p_{\text{out}}^{(2)} = 0.05$. We consider values of $\lambda \in \{10^{-3}, 10^{-2}, 10^{-1}, 10^0, 10^1, 10^2, 10^3\}$, different amount of labeled nodes $\{1\%, \ldots, 50\%\}$. We sample five random multilayer graphs with the corresponding parameters and 5 random samples of labeled nodes with a fixed percentage, and present the average classification error. In Fig. 7 we can see that in general the larger the value of $\lambda$ the smaller the classification error. In particular we can see that the performance does not present any relevant changes with $\lambda \leq 10^{-1}$.

**Experiments with real world datasets**. We analyze the effect of the regularization parameter $\lambda$ with real world datasets considered in Section 6. For each dataset we build the corresponding layer adjacency matrices by the taking symmetric $k$-nearest neighbour graph and take as similarity measure the Pearson linear correlation, (i.e. we take the $k$ neighbours with highest correlation), and take the unweighted version of it.

We fix nearest neighbourhood size to $k = 10$ and generate 10 samples of labeled nodes, where the percentage of labeled nodes per class is in the range $\{1\%, 2\%, \ldots, 25\%\}$. The average test errors are presented in Fig. 8, for power mean Laplacian regularizers $L_{-1}, L_{-2}, L_{-5}$, and $L_{-10}$. We can see that in general the best performance, i.e. smallest mean test classificaton error corresponds to values of $\lambda = 10, 10^2, 10^3$, verifying the choice of $\lambda = 10$ presented in Section 6. Moreover, we can see that the mean test error in general decreases with larger amounts of labeled data, which verifies our previous experiments on multilayer graphs following the Multilayer Stochastic Block Model.

(a1) $L_{-1}$      (a2) $L_{-2}$      (a3) $L_{-5}$      (a4) $L_{-10}$

(a) Dataset: 3sources

(b1) $L_{-1}$      (b2) $L_{-2}$      (b3) $L_{-5}$      (b4) $L_{-10}$

(b) Dataset: BBC

(c1) $L_{-1}$      (c2) $L_{-2}$      (c3) $L_{-5}$      (c4) $L_{-10}$

(c) Dataset: BBCS

(d1) $L_{-1}$      (d2) $L_{-2}$      (d3) $L_{-5}$      (d4) $L_{-10}$

(d) Dataset: Wikipedia

(e1) $L_{-1}$      (e2) $L_{-2}$      (e3) $L_{-5}$      (e4) $L_{-10}$

(e) Dataset: UCI

(f1) $L_{-1}$  (f2) $L_{-2}$  (f3) $L_{-5}$  (f4) $L_{-10}$

(f) Dataset: Citeseer

(g1) $L_{-1}$  (g2) $L_{-2}$  (g3) $L_{-5}$  (g4) $L_{-10}$

(g) Dataset: Cora

(h1) $L_{-1}$  (h2) $L_{-2}$  (h3) $L_{-5}$  (h4) $L_{-10}$

(h) Dataset: WebKB

Figure 8: Mean test classification error on real world datasets for different values of $\lambda$. Details in Sec. J.