[Reviews · NeurIPS 2019]

Reviewer 1



I think this paper is well prepared. I have the following comments. (1) I think the most important contribution is in theoretical aspect. Nevertheless, I cannot judge the proposed results is based on the previous works or these results in developed by the authors themselves. Thus, I suggest the authors to highlight their unique contributions. (2) I notice one assumption is that k clusters have equal size. I think this is too strict for real application. Can the authors give more insights about the results when this assumption does not hold? (3) As stated in Section 5, the authors have also touched the computational issue. I have not found some numerical comparison. I suggest the authors to added some numerical comparisons.

Reviewer 2



Originality: The paper is an extension of [1] to apply the Power Mean Laplacian to semi-supervised learning. The paper provides some insights about the new algorithm for the choice of p, depending on the noise assumption on the graphs. Quality: The theorems in the paper is insightful for understanding the algorithm. The provided numerical analysis is also convincing to demonstrate the theoretical results. The proposed method for solving the linear system efficiently also makes the algorithm more practical. However, I’m not sure if the empirical results were convincing enough as the authors claimed they didn’t tune to the parameters for all algorithms. Some of the baselines gives pretty terrible results, e.g. SMACD, which contradicts with the original paper. Clarity: The paper was written clearly. I can follow the paper pretty easily. There are a few typos I found in the paper: Page 2, line 71, “Moreover, not that” should be “note that” Page 5, line 183, “its overal performance is larger than the proposed approach”. overal -> overall, performance -> error? Significance: The proposed algorithm seems an interesting direction for semi-supervise learning. Even though the paper provides some efficient method for solving the linear system, the complexity of the algorithm was still pretty high, which might hinder the wide adoption of the algorithm. [1] A. Subramanya and P. P. Talukdar. Graph-Based Semi-Supervised Learning. Morgan & Claypool Publishers, 2014.

Reviewer 3



The paper discusses how to solve semi-supervised learning with multi-layer graphs. For single-layer graphs, this is achieved by label regression regularized by Laplacian matrix. For multi-layer, the paper argues that it should use a power mean Laplacian instead of the plain additive sum of Laplacians in each layer. This generalizes prior work including using the harmonic means. Some theoretical discussions follow under the assumptions from Multilayer Stochastic Block Model (MSBM), showing that specificity and robustness trade-offs can be achieved by adjusting the power parameter. I am not fully understanding or convinced by the Theorem 1. Particularly, it claims zero test error under stochastic models, which is either wrong or suggesting that the method may be impractical. Further, Corollary 1 and most of the experimental results support a negative norm parameter, which is rarely seen. I also checked the additional descriptions on Krylov subspace methods. Krylov subspace methods are important for sparse matrix factorization. However, I found no original contributions on this part from the authors, besides a (nice) simple mentioning. === Post rebuttal === I re-checked the paper. I agree that the paper seems intuitive and it makes incremental contributions. I will adjust my score to weak accept, with the following notes: The main contribution is a SSL method that is "provably robust against noise ... as long as one layer is informative and remaining layers are potentially just noise" as a limiting case. The intuition is that when p -> -inf, the smallest eigenvalues of the informative layers will dominate the other presumably larger eigenvalues of the noisy layers. The intuition is further extended to a series of consistency theorems using graph Laplacians in expectation, which is a limiting case given infinite amount of edge observations. While the intuition for p -> -infty is obvious, I would love to see more discussions around p = -1 case. The p=-1 case is included in Theorem 1, but not discussed in Corollary 1. Particularly, the assumption that "one layer is informative and the remaining layers are potentially just noise" no longer holds. Neither is p=-1 discussed in Section 4.2, because Figure 4 shows equally good results with p from -10 to 1, i.e., not specific to p=-1. The main evidence for p=-1 case is from the real-data experiments, but with insufficient discussions. The Krylov subspace methods need to be presented more clearly. Does it fundamentally change the orders of complexity or not?

[Author Response · NeurIPS 2019]

*We thank all reviewers. We think the negative impression of R5 is due to misunderstandings which we clarify.*

**R1:Highlight contributions.** We briefly summarize our contributions:

1. (Multilayer Graph Regularizer) We introduce a novel regularizer based on a one parameter family of Generalized
Matrix Power Means ($L_p$) for multilayer-graph based semi-supervised learning,

2. (Theorem 1 / Corollary 1 / Corollary 2) We provide conditions for zero classification error under a Multilayer
Stochastic Block Model in expectation. In particular we show that limit cases $p \to \infty$ and $p \to -\infty$ present different
robustness properties, e.g. $p \to -\infty$ is robust against noisy layers,

3. (Theorem 3) We present a novel alternative to Class Mass Normalization which provably performs well under the
Multilayer Stochastic Block Model,

4. (Complementary Layers) We show numerically that our approach is able to merge information when individual
layers present information about only one class but taking all together provides information about all classes,

5. (Numerical Scheme) We present a novel numerical scheme that shows that our approach is scalable to large multilayer
graphs. To our knowledge, this is the first numerical scheme for matrix functions like the matrix power means.

**R1:Case where clusters have different sizes.** We are currently working on provable properties for this case. We would
like to highlight that the case where in expectation all clusters are of the same size is a common case for graph-based
analysis on graph generative models like the stochastic block model (for example [11,12,26])

**R1:More comparisons of numerical scheme.** Fig. 1 depicts the re-
quested time-execution comparison on graphs of sizes $[0.5, 1, 2, 4, 8] \times 10^4$.
Observe that our matrix free approach for $L_{-1}$ (solid blue curve) is com-
petitive to state of the art approaches. In particular we see that:

1. $L_{-1}(ours)$ (our matrix free approach for $L_{-1}$) performs similarly as
TSS[28], outperforming AGML[23] and SMACD[9].

2. The fastest are ZooBP[6]/CGL[1]/TLMV[33] whose classification
performance is outperformed by ours (See Fig 1, Fig. 3 and Table 2)

**R4:Numerical Experiments (parameter setting).** We fix the parameters
of each method following the corresponding references as for SMACD
due to the high computational cost parameter tuning is unfeasible. An
analysis of the regularization parameter on our approach is available in
the supplementary material (Fig.6, page 23).

Figure 1: Time execution comparison.

**R4:Performance of SMACD.** We used the code provided by the authors (github.com/egujr001/SMACD) which has as
default parameter $\lambda = 0.1$. To address R4 concerns we performed the following efforts: **1)** Following[9] we explored
the regularization parameter $\lambda \in [10^{-8}, 10^6]$, **2)** We emailed the authors who suggested to remove possible self-loops,
and **3)** We consider all possible label permutations on training set to identify the optimal labeling. None of them
significantly improved the performance. Upon request we can omit it from the final version.

**R5:"This generalizes prior work using the harmonic means".** To our knowledge this is the first time that matrix
power means are considered as a regularizer for multilayer graph-based semi-supervised learning. It seems that R5
mixes up the single-layer approach of [35] which is based on the notion of harmonic functions with our Matrix Harmonic
Mean of Laplacians ($p = -1$). Apart from the word "harmonic" the two approaches are completely unrelated.

**R5:"Theorem 1 / Corollary 1 - is wrong or impractical".** We have the impression that R5 has missed that Theo-
rem 1/Corollary 1 are stated for the multilayer SBM in *expectation*. We don't see what is impractical. Claims that a
result is wrong require mathematical arguments (the proofs are in Sections B and C of the supplementary material).

**R5:"Corollary 1 - support a negative norm parameter, which is rarely seen".** We have the impression that there
is a misunderstanding. We are not considering any sort of "negative norm parameter": what we propose is a matrix
regularizer, i.e. the matrix power mean Laplacian with parameter $p$ (i.e. $L_p$), e.g. $L_{-1}$ is the matrix harmonic mean.
The corresponding optimization problem is (see Eq.1) $min_{f \in \mathbb{R}^n} \|f - Y\|_2^2 + \lambda f^T L_p f$, with solution $(I + \lambda L_p)f = Y$.

**R5: "no original contributions on Krylov Subspace methods".** We know the literature in numerical linear algebra
quite well. Our scheme is based on Krylov subspace solvers (PCG,GMRES[25]) together with Quadrature methods[10].
To our knowledge this is the first matrix-free approach for Matrix Power Means: most of the theory of matrix functions
considers functions of a single matrix (see[10]) whereas in our case we have a function of several matrices.

**R5:SBM is too easy.** The SBM is the most popular graph model [11,12,22,26]. We provide theoretical results based on
SBM in expectation, and show a competitive performance in real world graphs that do not follow the SBM (Section 6).

**R5:"I do not think that is the power mean that matters but ... spectral functions that are different from sum of
squares'- lot of work in this area".** We would be very grateful to R5 for references on this topic for multilayer graphs
- ideally some which also include theoretical guarantees.

[Meta-Review · NeurIPS 2019]

This paper makes a contribution toward the theory of semi-supervised learning for graph classification, as well as an efficient algorithm for computing the proposed classifier. This is an interesting problem and the reviewers agree the contribution is at least incremental. I suggest the authors carefully revise the paper to address reviewer concerns to get the maximum impact.